# Layer-wise Update Aggregation with Recycling for Communication-Efficient Federated Learning

**Jisoo Kim[1], Sungmin Kang[2], Sunwoo Lee[1]***
[1]Inha University
Incheon, Republic of Korea
[2]University of Southern California
Los Angeles, CA, USA
`starprin3@inha.edu, kangsung@usc.edu, sunwool@inha.ac.kr`

## Abstract

Expensive communication cost is a common performance bottleneck in Federated Learning (FL), which makes it less appealing in real-world applications. Many communication-efficient FL methods focus on discarding a part of model updates mostly based on gradient magnitude. In this study, we find that recycling previous updates, rather than simply dropping them, more effectively reduces the communication cost while maintaining FL performance. We propose `FedLUAR`, a Layer-wise Update Aggregation with Recycling scheme for communication-efficient FL. We first define a useful metric that quantifies the extent to which the aggregated gradients influence the model parameter values in each layer. `FedLUAR` selects a few layers based on the metric and recycles their previous updates on the server side. Our extensive empirical study demonstrates that the update recycling scheme significantly reduces the communication cost while maintaining model accuracy. For example, our method achieves nearly the same AG News accuracy as FedAvg, while reducing the communication cost to just 17%.

## 1 Introduction

While Federated Learning has become a distributed learning method of choice recently, there still exists a huge gap between practical efficacy and theoretical performance. Especially, the communication cost of model aggregation is one of the most challenging issues in realistic FL environments. It is well known that larger models exhibit stronger learning capabilities. The larger the model, the higher the communication cost. Thus, addressing the communication cost issue is crucial for realizing scalable and practical FL applications.

Many communication-efficient FL methods focus on partially 'dropping' model parameters and thus their updates. Quantization-based FL methods reduce communication costs by lowering the numerical precision of transmitted model parameters, representing each parameter with a lower bit-width. Pruning-based FL methods directly remove a portion of model parameters to avoid the associated gradient computations and communication overhead. Model reparameterization-based FL methods adjust the model architecture using matrix decomposition techniques, reducing the total number of parameters.While all these approaches reduce the communication cost, they commonly compromise learning capability by either reducing the number of parameters or degrading the data representation quality.

In this paper, we propose `FedLUAR`, a Layer-wise Update Aggregation with Recycling method for communication-efficient and accurate FL. Instead of dropping the updates for a part of model

---

*Corresponding Author

39th Conference on Neural Information Processing Systems (NeurIPS 2025).

parameters, we consider 'reusing' the old updates multiple times in a layer-wise manner. Our study first defines a useful metric that quantifies the extent to which the aggregated gradients influence the model parameter values in each layer. Based on the metric, a small number of layers are selected to recycle their previous updates on the server side. Clients can omit these updates when sending their locally accumulated updates to the server. This layer-wise update recycling method allows only a subset of less important layers to lose their update quality while maintaining high-quality updates for all the other layers. Our study shows that, by carefully choosing the update recycling layers, the model aggregation cost can be dramatically reduced while maintaining the model accuracy.

Our study provides critical insights into achieving a practical trade-off between communication cost reduction and the level of noise introduced by any types of communication-efficient FL methods. First, by introducing noise into layers where the update magnitude is small relative to the model parameter magnitude, the adverse impact of the noise can be minimized, thus preserving FL performance. Second, our study empirically demonstrates that the update recycling approach achieves faster loss convergence compared to simply dropping updates for the same layers. Our theoretical analysis also shows that `FedLUAR` converges to a neighborhood of a stationary point when updates are recycled in a sufficiently small number of layers. We designed our FL method to leverage these findings and it can be readily applied to various FL applications to improve scalability.

We evaluate the performance of `FedLUAR` [2] using representative benchmark datasets: CIFAR-10 [19], CIFAR-100, FEMNIST [4], and AG News [42] We first compare `FedLUAR` to several state-of-the-art communication-efficient FL methods: Look-back Gradient Multiplier [2], FedPAQ [31], FedPara [12], PruneFL [13], FedDropoutAvg [8], and FedBAT [23]. We also compare the performance between using and not using the recycling method for advanced FL optimizers. Finally, we provide extensive ablation study results that further validate the efficacy of our proposed method, including performance comparisons based on the number of layers with recycled updates. These experimental results and our analysis show that `FedLUAR` provides a novel and efficient approach to reducing the communication cost while maintaining the model accuracy in FL environments.

## 2 Related Work

**Structured Model Compression** – Several low-rank decomposition-based FL methods have been proposed, which re-parameterize the model weights to reduce either computational or communication costs [12, 28, 35]. These methods modify the model architecture in a structured way using various tensor approximation techniques [18]. The re-parameterization methods often increase the number of network layers, resulting in higher implementation complexity and higher computational costs. Moreover, they struggle to maintain model performance when the rank is significantly reduced.

**Sketched Model Compression** – Quantization-based FL methods have been actively studied to reduce the number of bits used per parameter [5, 9, 31, 36]. Model pruning methods, such as PruneFL [13], FedMP [14], FedPruning [22], GossipFL [33], and SpaFL [17], remove a portion of model parameters to reduce both computational and communication costs. These methods are categorized under a *sketching* approach. Although quantization methods reduce communication overhead, they uniformly degrade the data representation quality of all parameters, overlooking their varying contributions to the training process. The pruning methods potentially harm the model's learning capability since they directly reduce the number of parameters.

**Other Communication-Efficient FL Methods** – FedLAMA [21] adaptively adjusts model aggregation frequency in a layer-wise manner. Dynamic model aggregation method proposed in [15] aggregates local models in a decentralized manner. FedKD [37] reduces communication cost by employing knowledge distillation in place of model aggregation. These methods address the high communication cost issue in FL. However, they do not consider the possibility of 'reusing' previous gradients. In this work, we focus on recycling previously computed gradients to reduce the communication cost. Bandwidth-aware Compression Ratio Scheduling (BCRS) [34] adjusts the top-k compression ratio in a network bandwith-aware manner. YOGA [26] adopts a layer-wise aggregation strategy based on layer priority, which shares conceptual similarities with our proposed method. However, it assumes a peer-to-peer decentralized FL environment without a central server, which is not applicable to server-based FL scenarios.

**Gradient-Weight Ratio in Deep Learning** – A few of the recent works focus on utilizing gradient-weight ratio. Some researchers adjust learning rate based on the ratio to improve the model per-

---

[2] https://github.com/swblaster/FedLUAR

formance [27, 39]. These previous works theoretically demonstrate that the gradient-weight ratio delivers useful insights that can be utilized to adjust the inherent noise scale of stochastic gradients. In this study, we propose and employ a similar metric in FL environments: the ratio of accumulated updates to the initial model parameters at each communication round.

**Gradient Dynamics** – Depending on the geometry of the parameter space, the gradient may remain consistent over several training iterations [2]. It has also been shown that the loss landscape becomes smoother as the batch size increases, and thus the stochastic gradients can remain similar for more iterations [16, 20, 25]. We explore the possibility of 'recycling' such stable gradients multiple times. By recycling previous updates, clients can avoid update aggregation in some network layers. In the following section, we will discuss how to safely recycle updates in a layer-wise manner.

## 3 Method

In this section, we first introduce a layer prioritization metric that can be efficiently calculated during training. Then, we present a communication-efficient FL method that recycles updates for layers with low priority. Finally, we provide a theoretical guarantee of convergence for the proposed FL method.

### 3.1 Motivation

Many existing communication-efficient FL methods focus on how to reduce the communication cost while keeping the gradient magnitude as close as possible to the original. For instance, update sparsification methods select a subset of parameters with small gradients and omit their updates [1, 3]. Layer-wise model aggregation methods selectively aggregate the local updates at a subset of layers with small gradient norms [2, 21]. These methods commonly assume that larger gradients indicate greater importance.

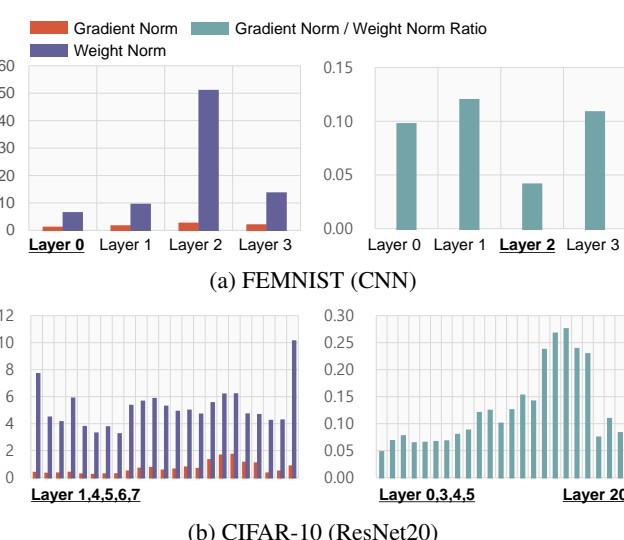

(a) FEMNIST (CNN)

(b) CIFAR-10 (ResNet20)

Figure 1: The layer-wise gradient norm and weight norm comparison (left) and the ratio of the gradient norm to the weight norm (right). It is clearly shown that the layers with the smallest gradients do not always least significantly affect the model parameter values.

Figure 1 shows layer-wise intermediate data collected from FEMNIST and CIFAR-10 training. The left-side chart compares the gradient norm and the weight norm while the right-side chart shows the ratio of the gradient norm to the weight norm. The IDs of a few layers with small gradient magnitudes (left) and ratios (right) are shown at the bottom of the charts. The detailed hyper-parameter settings can be found in Appendix.

The key message from this empirical study is that layers with small gradients do not necessarily show a small ratio of gradient norm to weight norm. The small ratio can be interpreted as a less significant impact of the update on changes in parameter value. Even when the gradient is large, if the corresponding parameter is also large, its effect on the layer's output will remain minimal. From the perspective of each layer, therefore, the ratio may serve as a more critical metric than the magnitude of the gradients alone. This observation motivates us to explore a novel approach to prioritizing network layers by focusing on the ratio of gradient magnitude to weight magnitude rather than solely monitoring gradient magnitude.

### 3.2 Layer-wise Update Recycling

**Gradient-Weight Ratio Analysis** – We prioritize network layers using the ratio of gradient magnitude to weight magnitude. Given $L$ layers of a neural network, the prioritization metric $s_{t,l}$ is defined as

---

**Algorithm 1** Layer-wise Update Aggregation with Recycling (LUAR)

---

1: **Input:** $\Delta_t^i$: the latest local updates, $\hat{\Delta}_{t-1}$: the updates used at the previous round, $\mathcal{R}_t$: the set of recycling layers, $\delta$: the number of recycling layers
2: Clients send out the local updates: $\mathbf{u}_t^i = [\Delta_{t,l}^i], \forall l \notin \mathcal{R}_t$
3: Server aggregates the updates: $\mathbf{u}_t = \frac{1}{a} \sum_{i=1}^a \mathbf{u}_t^i$
4: $\mathbf{r}_t = [\hat{\Delta}_{t-1,l}], \forall l \in \mathcal{R}_t$
5: $\hat{\Delta}_t = [\mathbf{r}_t, \mathbf{u}_t]$
6: Update the recycling scores: $\mathbf{s}_{t,l}$            ▷ Eq. (1)
7: $\mathbf{p}^t \leftarrow$ Calculate $p_l^t, \forall l \in [L]$            ▷ Eq. (2)
8: $\mathcal{R}_{t+1} \leftarrow$ Random_Choice($[L], \delta, \mathbf{p}^t$)
9: **Output:** $\hat{\Delta}_t, \mathcal{R}_{t+1}$

---

follows.

$$s_{t,l} = \frac{\|\Delta_{t,l}\|}{\|\mathbf{x}_{t,l}\|}, \forall l \in \{0, \cdots, L-1\} \tag{1}$$

where $\Delta_{t,l}$ is the accumulated local updates averaged across all the clients at round $t$ for layer $l$, and $\mathbf{x}_{t,l}$ is the initial model parameters of layer $l$ at round $t$. Intuitively, this metric quantifies the relative gradient magnitude based on parameter magnitude. If $s_{t,l}$ is measured large, we expect the layer's parameters to move fast in the parameter space making it sensitive to the update correctness. In contrast, if $s_{t,l}$ is small, the layer's parameters will not be dramatically changed after each update. We assign low priority to layer $l$ if its $s_{t,l}$ is small, and high priority if it is large. In this way, all the $L$ layers can be prioritized based on how actively the parameters are changed after each round.

This metric can be efficiently measured on the server-side. The $\mathbf{x}_{t,l}$ is already stored on the server before every communication round. All FL methods aggregate the local updates after every round, and thus $\Delta_{t,l}$ is also already ready to be used on the server. Therefore, $s_{t,l}$ can be easily measured without any extra communications. This is a critical advantage considering the limited network bandwidth in typical FL environments.

**Layer-wise Stochastic Update Recycling Method** – We design a novel FL method that recycles the previous updates for a subset of layers. The first step is to calculate a probability distribution of $L$ network layers based on the prioritization metric shown in (1). The probability of layer $l$ to be chosen is computed as follows:

$$p_{t,l} = \frac{1/s_{t,l}}{\sum_{l=0}^{L-1} 1/s_{t,l}}, \forall l \in \{0, \cdots, L-1\}. \tag{2}$$

Each layer has a weight factor $\frac{1}{s_{t,l}}$ so that it is less likely chosen if its priority is low. Dividing it by $\sum_{i=0}^{L-1} 1/s_{t,l}$ ensures the sum of all weight factors equals 1, allowing $p$ values to be directly used as a weight factor of random sampling. Second, our method randomly samples $\delta$ layers using the probability distribution $\mathbf{p}$ shown in (2). We define those sampled layers at round $t$ as $\mathcal{R}_t$. Finally, the sampled $\delta$ layers are updated using the previous round's updates instead of the latest updates. That is, the clients do not send to the server the local updates for those $\delta$ layers.

It is worth noting that the weighted random sampling-based layer selection prevents the updates for low-priority layers from being recycled excessively. When low-priority layers are not sampled, their updates will be normally aggregated on the server-side and thus their $s_{t,l}$ values can be updated. We will analyze the impact of this stochastic layer selection scheme on the overall performance of the update recycling method in Section 4.

We formally define the update recycling method as follows.

$$\mathbf{u}_t = [\Delta_{t,l}], \forall l \notin \mathcal{R}_t \tag{3}$$

$$\mathbf{r}_t = [\hat{\Delta}_{t-1,l}], \forall l \in \mathcal{R}_t \tag{4}$$

$$\hat{\Delta}_t = [\mathbf{r}_t, \mathbf{u}_t], \tag{5}$$

where $\mathbf{u}_t$ is the updates for layers not included in $\mathcal{R}_t$, $\mathbf{r}_t$ is the recycled updates for $\delta$ layers in $\mathcal{R}_t$, and $\hat{\Delta}_t$ is the global update composed of $\mathbf{u}_t$ and $\mathbf{r}_t$. Algorithm 1 shows the Layer-wise Update

---

**Algorithm 2** Federated Learning with Layer-wise Update Aggregation with Recycling (`FedLUAR`)

---

1: **Input:** $a$: the number of active clients per round, $T$: the total number of rounds
2: $\mathcal{R}_0 \leftarrow$ an empty set.
3: **for** $t \in \{0, \cdots, T-1\}$ **do**
4:     $\mathcal{A} = \text{Random\_Choice}([\mathcal{N}], a)$.
5:     Server sends out $\mathbf{x}_t, \mathcal{R}_t$ to the clients $\forall i \in [\mathcal{A}]$.
6:     Client receives the model: $\mathbf{x}_{t,0}^i = \mathbf{x}_t$.
7:     **for** $j \in \{1, \cdots, \tau\}$ **do**
8:         $\mathbf{x}_{t,j}^i = \text{Local\_Update}(\mathbf{x}_{t,j-1}^i)$.
9:     **end for**
10:    Clients calculate the update $\Delta_t^i = \mathbf{x}_{t,\tau}^i - \mathbf{x}_{t,0}^i$.
11:    $\hat{\Delta}_t, \mathcal{R}_{t+1} = \text{LUAR}(\Delta_t^i, \hat{\Delta}_{t-1}, \mathcal{R}_t)$                                      ▷ Alg.1
12:    $\mathbf{x}_{t+1} = \mathbf{x}_t + \hat{\Delta}_t$
13: **end for**
14: **Output:** $\mathbf{x}_T$

---

Aggregation with Recycling (LUAR) method. Note that the number of layers whose update will be recycled, $\delta$, is a user-tunable hyper-parameter. We will further discuss how $\delta$ affects the model accuracy as well as the communication cost in Appendix A.4.

**Federated Learning Framework** – Algorithm 2 shows `FedLUAR`, a FL framework built upon LUAR (Alg. 1). Before the active clients download the initial model parameters $\mathbf{x}_t$, the server informs them of the set of layers to be recycled, denoted by $\mathcal{R}_t$ (line 5). Subsequently, each client performs local training for $\tau$ iterations. The resulting local updates are then aggregated using LUAR (line 11). Finally, the global model is updated using $\hat{\Delta}_t$. Figure 2 shows a schematic illustration of `FedLUAR`. Each client transmits updates only for the layers with large $s_{t,l}$ values. For the remaining layers, the server recycles the previous updates. When $\delta = 0$, $\hat{\Delta}_t$ becomes the same as $\Delta_t$, effectively reducing the method to vanilla FedAvg. In this paper, we use FedAvg as the base federated optimization algorithm for simplicity. However, extending it to more advanced FL optimizers is straightforward, as LUAR is agnostic to the choice of optimizer.

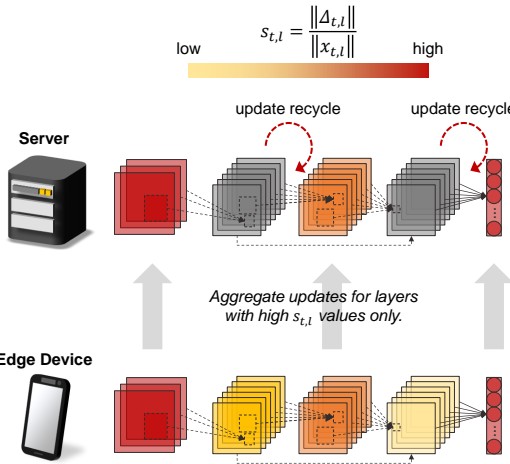

Figure 2: Schematic illustration of `FedLUAR`.

**Potential Limitations** – `FedLUAR` requires the server to notify active clients about which layer updates should be omitted when uploading their local updates. This introduces an additional communication cost compared to other FL methods. However, this extra overhead is likely negligible, as the list of recycled layer IDs ($\delta$ integers) can be transmitted along with the initial model parameters at the beginning of each communication round.

### 3.3 Theoretical Analysis

We consider non-convex and smooth optimization problems:

$$\min_{\mathbf{x} \in \mathbb{R}^d} F(\mathbf{x}) := \frac{1}{m} \sum_{i=1}^m F_i(\mathbf{x}),$$

where $F_i(\mathbf{x}) = \mathbb{E}_{\xi_i \sim D_i}[f(\mathbf{x}, \xi_i)]$ is the local loss function associated with the local data distribution $D_i$ of client $i$ and $m$ is the number of clients. Our analysis is based on the following assumptions.
**Assumption 1.** *(Lipschitz continuity) There exists a constant $\mathcal{L} > 0$, such that $\|\nabla F_i(\mathbf{x}) - \nabla F_i(\mathbf{y})\| \le \mathcal{L}\|\mathbf{x} - \mathbf{y}\|, \forall \mathbf{x}, \mathbf{y} \in \mathbb{R}^d$, and $i \in [m]$.*

**Assumption 2.** *(Unbiased local gradients) The local gradient estimator is unbiased such that $\mathbb{E}_{\xi_i \sim D_i}[\nabla f(\mathbf{x}, \xi_i)] = \nabla F_i(\mathbf{x}), \forall i \in [m]$.*

**Assumption 3.** *(Bounded local and global variance) There exist two constants $\sigma_L > 0$ and $\sigma_G > 0$, such that the local gradient variance is bounded by $\mathbb{E}[\|\nabla f(\mathbf{x}, \xi_i) - \nabla F_i(\mathbf{x})\|]^2 \le \sigma_L^2, \forall i \in [m]$, and the global variability is bounded by $\mathbb{E}\left[\|\nabla F_i(\mathbf{x}) - \nabla F(\mathbf{x})\|^2\right] \le \sigma_G^2, \forall i \in [m]$.*

**Noise Definition** – We define $\hat{g}$ as a stochastic gradient vector that corresponds to the $\delta$ layers whose updates will be recycled. Likewise, the corresponding full-batch gradient is defined as $\nabla \hat{F}(\mathbf{x})$. We quantify the ratio of $\|\nabla \hat{F}(\mathbf{x}_t)\|^2$ to $\|\nabla F(\mathbf{x}_t)\|^2$, which is $\le 1$, as $\kappa$. By the definition of $\nabla \hat{F}(\mathbf{x})$, $\kappa$ goes to zero if none of the layers recycle their updates. To analyze the impact of the update recycling in Algorithm 1, we also define the quantity of noise $n_t$ as follows.

$$n_t := \hat{\Delta}_t - \Delta_t = \frac{1}{m} \sum_{i=1}^{m} \sum_{j=0}^{\tau-1} \left(\hat{g}_{t-k,j}^i - \hat{g}_{t,j}^i\right). \tag{6}$$

The $k$ in (6) represents the degree of update staleness, which increases as the update is recycled in consecutive communication rounds. As shown in Algorithm 1, our proposed method does not specify the upper bound of $k$ and adaptively selects the recycling layers based only on $s_{t,l}$ values. Therefore, we analyze the convergence rate of Algorithm 2 without any assumptions on the $k$ value.

Herein, we analyze the convergence rate of Algorithm 2 (See Appendix for proofs).

**Lemma 1.** *(noise) Under assumption $1 \sim 3$, if the learning rate $\eta \le \frac{1}{\mathcal{L}\tau}$, the accumulated noise is bounded as follows.*

$$\sum_{t=0}^{T-1} \mathbb{E}\left[\|n_t\|^2\right] \le 4T\tau^2\sigma_L^2 + 8T\tau^2\sigma_G^2$$

$$+ 8\kappa\tau^2 \sum_{t=0}^{T-1} \mathbb{E}\left[\|\nabla F(\mathbf{x}_t)\|^2\right]$$

$$+ \frac{8\tau L^2}{m} \sum_{t=0}^{T-1} \sum_{i=1}^{m} \sum_{j=0}^{\tau-1} \mathbb{E}\left[\|\mathbf{x}_{t,j}^i - \mathbf{x}_t\|^2\right],$$

*where $m$ is the number of clients.*

**Theorem 2.** *Under assumption $1 \sim 3$, if the learning rate $\eta \le \frac{1-16\kappa}{6\sqrt{30}\mathcal{L}\tau}$ and $\kappa < \frac{1}{16}$, we have*

$$\sum_{t=0}^{T-1} \mathbb{E}\left[\|\nabla F(\mathbf{x}_t)\|^2\right] \le \frac{4}{(1-16\kappa)\eta\tau}\left(F(\mathbf{x}_0) - F(\mathbf{x}_T)\right)$$

$$+ \frac{4T}{1-16\kappa}\left(\frac{\mathcal{L}\eta}{m} + 4 + 9\mathcal{L}^2\right)\sigma_L^2 \tag{7}$$

$$+ \frac{1080T\mathcal{L}^2\eta^2\tau^2}{1-16\kappa}\sigma_G^2.$$

**Remark 1.** *Lemma 1 shows that the update recycling method ensures the noise magnitude bounded regardless of how many times the updates are recycled and how many layers recycle their updates if $\kappa$ is sufficiently small. This result can serve as a foundation that allows users to safely recycle updates in a layer-wise manner, thereby reducing communication costs.*

**Remark 2.** *Algorithm 2 converges to a neighborhood of a stationary point, as the second and third terms on the right-hand side of (7) do not vanish as $T \to \infty$. Furthermore, the term of $(4 + 9\mathcal{L}^2)\sigma_L^2$ is independent of the learning rate $\eta$ and remains non-zero even as $\eta \to 0$. Although it does not ensure convergence to an exact minimum, this rough guarantee is considered useful in real-world applications [7, 41].*

In general, as the degree of non-IIDness increases, the global variance tends to grow due to greater discrepancies among local datasets. As shown in the final term on the right-hand side of (7), recycling updates in more layers increases the coefficient $\frac{1080T\mathcal{L}^2\eta^2\tau^2}{1-16\kappa}$, which in turn amplifies the final term. Consequently, the model is expected to converge more slowly. This suggests using smaller learning rate as the degree of non-IIDness increases to maintain the convergence rate.

## 3.4 Memory Usage Analysis

In FedAvg, server should receive local models from all active clients. Consequently, the maximum memory footprint is $a \cdot d$, where $a$ is the number of active clients and $d$ is the model size. By contrast, `FedLUAR` receives local models from active clients except $\delta$ layers. Thus, the memory footprint is $a \cdot (d - k)$, where $k$ is the size of $\delta$ layers. Instead, the previous global update should be kept in the memory space for the $\delta$ layers, consuming $k$ space only. Therefore, `FedLUAR`'s memory footprint is $a \cdot (d - k) + k < a \cdot d$.

To support our analysis, we actually measured the memory footprint of FedAvg and `FedLUAR` during training. First, the total number of clients

| Dataset (Model) | Algorithm | $\delta$ | Memory Footprint (MB) |
|---|---|---|---|
| CIFAR-10 (ResNet20) | FedAvg | - | 33.49 |
| | `FedLUAR` | 10 | 15.23 |
| CIFAR-100 (WRN28-10) | FedAvg | - | 4,462.80 |
| | `FedLUAR` | 14 | 2,604.88 |
| FEMNIST (CNN) | FedAvg | - | 806.11 |
| | `FedLUAR` | 2 | 204.73 |
| AG News (DistillBERT) | FedAvg | - | 8,294.18 |
| | `FedLUAR` | 30 | 1,825.42 |

Table 1: Comparison of memory usage observed during training.

is 128 and only randomly selected 32 clients are activated at each communication round. We use MPI to run FL on 2 GPUs. Thus, each process locally train 16 models and then all the locally trained models are aggregated using `MPI_Allreduce()`. Table 1 shows the memory footprint of each process, observed during training under this setting. We can clearly see that `FedLUAR` uses less memory space than FedAvg. This advantage is directly related to the reduced communication cost, which will be discussed in Section 4.3.

## 4 Experiments

**Experimental Settings** – All experiments are conducted on a GPU cluster which has 2 NVIDIA A6000 GPUs per machine. We use TensorFlow 2.15.0 for training and MPI for model aggregations. All experiments were performed at least 3 times, and the average accuracies are reported.

**Datasets** – We evaluate the performance of our proposed method on representative benchmarks: CIFAR-10 (ResNet20 [10]), CIFAR-100 (Wide-ResNet28 [40]), FEMNIST (CNN), and AG News (DistillBert [32]). When tuning hyper-parameters, we conduct a grid search with a sufficiently small unit size (e.g., 0.1 for learning rate). To generate non-IID datasets, we use label-based Dirichlet distributions with $\alpha = 0.1$, which indicates highly non-IID conditions.

**Data Heterogeneity** – For IID datasets, we simulate non-IID settings using Dirichlet distributions. The concentration coefficient $\alpha$ is set to 0.1 for CIFAR-10/100 and 0.5 for AG News.

### 4.1 Comparative Study

We first present an accuracy comparison among SOTA communication-efficient FL methods below.

- LBGM (Low-rank Approximation) [2]
- FedPAQ (Quantization) [31]
- FedPara (Reparameterization) [12]
- PruneFL (Pruning) [13]
- FedDropoutAvg (Dropping) [8]
- FedBAT (Binarization) [23]

Table 2 shows the performance comparison (See Appendix for the detailed settings). The total number of clients is 128 and randomly chosen 32 clients participate in every communication round. Note that the FL methods cannot have exactly the same communication cost due to differences in their mechanisms. To ensure fair comparisons, we find algorithm-specific settings that achieve accuracy reasonably close to the baseline (FedAvg) while minimizing communication costs, and then compare the validation accuracy across algorithms.

Overall, `FedLUAR` achieves accuracy comparable to the baseline while significantly reducing communication costs across all four benchmarks. Notably, for FEMNIST and AG News, it matches FedAvg's accuracy with less than $20\%$ of the communication cost. Our method also outperforms all other SOTA methods. While FedPAQ and FedBAT reduce communication cost, they suffer from noticeable accuracy drops. Regardless of the dataset, `FedLUAR` consistently delivers the highest accuracy among communication-efficient FL methods. These results demonstrate that `LUAR` effectively finds less critical layers and recycles their updates, minimizing the communication cost without sacrificing performance.

| Method | CIFAR-10 (ResNet20) | | CIFAR-100 (WRN-28) | | FEMNIST (CNN) | | AG News (DistillBERT) | |
|--------|----------|------|----------|------|----------|------|----------|------|
| | Accuracy | Comm | Accuracy | Comm | Accuracy | Comm | Accuracy | Comm |
| FedAvg | $61.27 \pm 0.7\%$ | 1.00 | $59.88 \pm 0.8\%$ | 1.00 | $71.01 \pm 0.4\%$ | 1.00 | $82.66 \pm 0.2\%$ | 1.00 |
| LBGM | $54.87 \pm 0.5\%$ | 0.65 | $57.13 \pm 0.2\%$ | 0.87 | $69.83 \pm 1.0\%$ | 0.71 | $77.96 \pm 0.1\%$ | 0.23 |
| FedPAQ | $57.42 \pm 0.2\%$ | 0.50 | $36.15 \pm 0.1\%$ | 0.50 | $71.54 \pm 0.1\%$ | 0.25 | $82.72 \pm 0.1\%$ | 0.25 |
| FedPara | $55.16 \pm 0.1\%$ | 0.51 | $46.14 \pm 0.1\%$ | 0.61 | $67.69 \pm 0.1\%$ | 0.22 | $75.22 \pm 0.1\%$ | 0.69 |
| PruneFL | $56.76 \pm 0.1\%$ | 0.51 | $59.40 \pm 0.1\%$ | 0.69 | $69.42 \pm 0.4\%$ | 0.19 | $77.25 \pm 0.1\%$ | 0.22 |
| FDA | $56.54 \pm 0.3\%$ | 0.50 | $51.25 \pm 0.1\%$ | 0.60 | $70.61 \pm 0.1\%$ | 0.25 | $64.94 \pm 0.1\%$ | 0.50 |
| FedBAT | $39.56 \pm 0.1\%$ | 0.03 | $47.24 \pm 0.1\%$ | 0.03 | $68.27 \pm 0.1\%$ | 0.03 | $76.38 \pm 0.1\%$ | 0.57 |
| FedLUAR | $\mathbf{60.15 \pm 0.7\%}$ | **0.47** | $\mathbf{59.73 \pm 0.6\%}$ | **0.61** | $\mathbf{73.17 \pm 0.1\%}$ | **0.18** | $\mathbf{82.80 \pm 0.1\%}$ | **0.17** |

Table 2: Classification performance comparison. Comm denotes the communication cost relative to FedAvg.

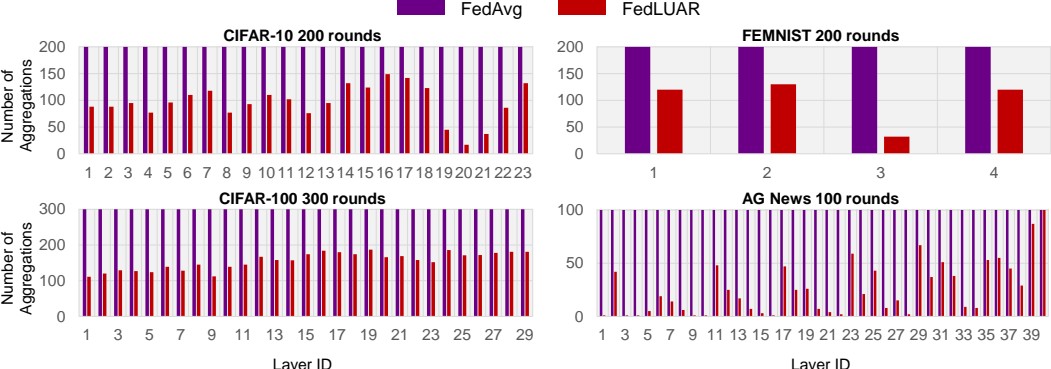

Figure 3: Number of model aggregations per layer. `FedLUAR` significantly reduces aggregation frequency across all benchmarks. The gap from FedAvg indicates how often updates were recycled (i.e., communications skipped).

## 4.2 Harmonization with Other FL Methods

The proposed FL method does not have any dependencies on the local training algorithm. To demonstrate this, we apply `LUAR` to several advanced FL methods, including FedProx [24], MOON [29], FedOpt [30], FedMut [11], FedACG [6], and PruneFL [13], and analyze its effect on model accuracy. Table 3 shows CIFAR-10 and FEMNIST accuracy comparisons (See Appendix for details). `LUAR` maintains the validation accuracy while significantly reducing the communication cost across all three benchmarks. Comm denotes communication cost relative to full model averaging. For instance, FedPAQ reduces communication to $50\%$ of FedAvg, while `LUAR` further reduces it to $22\%$ of FedPAQ, just $11\%$ of FedAvg, without compromising accuracy. These results show that `LUAR` can effectively complement advanced FL algorithms and is readily applicable to real-world scenarios.

## 4.3 Communication Cost Analysis

`FedLUAR` enables clients to skip uploading updates for less important layers, thereby reducing communication costs. Figure 3 shows the number of communications for each layer. The communication count charts indicate that `FedLUAR` requires significantly fewer communications than vanilla FedAvg, while achieving comparable model accuracy. An interesting observation is that, in FEMNIST and AG News, the layer with the largest number of parameters tends to be recycled most frequently, resulting in a substantial reduction in total communication cost. However, this trend is not observed in the CIFAR-10 and CIFAR-100 benchmarks. Thus, we conclude that the proposed method is independent of layer size and specific model architectures, as it adaptively identifies the least significant layers regardless of the model design.

## 4.4 Ablation Study

To further validate the effectiveness of the proposed metric shown in (1), we conduct an ablation study as follows. Fixing the number of layers to recycle updates, $\delta$, we measure model accuracy using different layer selection metrics. By comparing these accuracies, we can determine which metric is most effective in identifying the least critical layers in terms of their contribution to the global

| | Periodic Averaging | LUAR (Proposed) | Comm | $\delta$ |
|---|---|---|---|---|
| FedProx | $61.74 \pm 0.1\%$ | $61.20 \pm 0.1\%$ | 0.54 | |
| FedPAQ | $57.42 \pm 0.2\%$ | $57.40 \pm 0.2\%$ | 0.33 | |
| FedOpt | $62.42 \pm 0.1\%$ | $62.28 \pm 0.2\%$ | 0.50 | |
| MOON | $62.33 \pm 1.2\%$ | $61.65 \pm 0.1\%$ | 0.51 | 10 |
| FedMut | $61.27 \pm 0.1\%$ | $60.42 \pm 0.1\%$ | 0.56 | |
| FedACG | $65.02 \pm 0.1\%$ | $64.28 \pm 0.1\%$ | 0.55 | |
| PruneFL | $56.76 \pm 0.1\%$ | $55.43 \pm 0.1\%$ | 0.49 | |

(a) CIFAR-10 (ResNet20)

| | Periodic Averaging | LUAR (Proposed) | Comm | $\delta$ |
|---|---|---|---|---|
| FedProx | $71.94 \pm 0.1\%$ | $73.45 \pm 0.1\%$ | 0.09 | |
| FedPAQ | $71.54 \pm 0.1\%$ | $71.15 \pm 0.1\%$ | 0.11 | |
| FedOpt | $72.34 \pm 0.1\%$ | $71.91 \pm 0.1\%$ | 0.22 | |
| MOON | $71.55 \pm 0.1\%$ | $71.63 \pm 0.1\%$ | 0.24 | 2 |
| FedMut | $71.91 \pm 0.1\%$ | $72.31 \pm 0.1\%$ | 0.26 | |
| FedACG | $72.16 \pm 0.1\%$ | $71.94 \pm 0.1\%$ | 0.21 | |
| PruneFL | $69.42 \pm 0.1\%$ | $69.11 \pm 0.1\%$ | 0.11 | |

(b) FEMNIST (CNN)

Table 3: CIFAR-10 and FEMNIST performance comparison between before and after applying LUAR. LUAR is applied to half of the model layers for both datasets, using ResNet20 for CIFAR-10 and a CNN for FEMNIST. The *Comm* column shows the ratio of LUAR's cost to the FedAvg's cost.

| Layer Selection Scheme | CIFAR-10 | | FEMNIST | | AG News | |
|---|---|---|---|---|---|---|
| | Acc. (%) | Comm. | Acc. (%) | Comm. | Acc. (%) | Comm. |
| Random | 53.94% | 0.48 | 71.10% | 0.51 | 80.27% | 0.23 |
| Top (input-side) | 56.03% | 0.73 | N/A | | 79.71% | 0.21 |
| Bottom (output-side) | 45.02% | 0.24 | 69.54% | 0.13 | 81.14% | 0.45 |
| Gradient norm | 55.88% | 0.55 | 70.91% | 0.70 | 75.06% | 0.22 |
| Deterministic recycling | 48.47% | **0.20** | 69.08% | **0.02** | 80.22% | **0.15** |
| LUAR(Proposed) | **60.15%** | 0.47 | **73.17%** | 0.18 | **82.80%** | 0.17 |

Table 4: Performance comparison with different layer selection schemes. For CIFAR-10 and FEMNIST, half the layers were reused; for AG News, 30 layers. *Comm.* denotes communication cost normalized to FedAvg. Selecting top layers in FEMNIST leads to early-stage divergence.

model training. In particular, we compare the classification performance between the most popular gradient-based layer selection and our proposed LUAR. Table 4 shows the performance comparisons.

This ablation study provides several key insights. First, LUAR outperforms uniform random sampling, demonstrating that our proposed metric (1) effectively identifies less critical layers. Second, even with the same metric, a deterministic selection strategy yields lower accuracy. Persistently recycling updates for layers with low $s_{t,l}$ values can cause them to be too much outdated, introducing excessive noise that degrades model performance. Third, LUAR consistently outperforms the gradient norm-based method, supporting our earlier observation (Fig. 1) that gradient magnitude alone is insufficient to assess update importance. We thus conclude that the gradient-to-weight ratio best captures update quality, achieving the highest accuracy while significantly reducing communication costs.

Additionally, we compare the classification performance of update dropping and recycling schemes. Many existing communication-efficient FL methods merely drop a subset of updates. Table 5 presents the performance comparisons. Here, *Dropping* refers to the case where the $\delta$ least critical layers are selected using LUAR and their updates are dropped instead of being recycled. As expected, *Dropping* achieves the same communication cost reduction as *Recycling*, but its accuracy is significantly lower than that of *Recycling*. This ablation study clearly demonstrates the superior performance of our proposed update recycling scheme.

| Dataset | Dropping | Recycling | Comm. Cost | $\delta$ |
|---|---|---|---|---|
| CIFAR-10 | $46.89 \pm 0.1\%$ | $50.07 \pm 1.6\%$ | 0.30 | 16 |
| FEMNIST | $64.69 \pm 0.2\%$ | $73.17 \pm 1.1\%$ | 0.18 | 2 |
| AG News | $77.05 \pm 0.1\%$ | $82.80 \pm 0.1\%$ | 0.17 | 30 |

Table 5: Benchmark performance comparison between update dropping and update recycling schemes.

**How much does it accelerate?** – Figure 4 shows the learning curves for CIFAR-10 and AG News. The x-axis represents the communication cost relative to FedAvg. To highlight the difference clearly, we selectively present comparisons among four methods only. The comparison clearly shows that FedLUAR achieves similar accuracy to FedAvg much faster than other SOTA communication-efficient FL methods. Since our method incurs little to no additional computational cost, the same performance gain can be expected in terms of the end-to-end training time in realistic FL environments. In our empirical study, we observed the same performance gains across many different FL benchmarks. See Appendix for more curve charts and the detailed experimental settings.

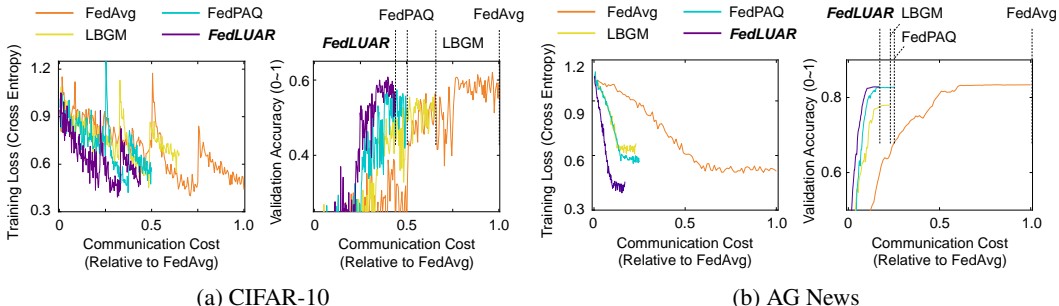

Figure 4: Learning curves for CIFAR-10 (ResNet20) and AG News (DistillBERT), with communication cost (x-axis) normalized to FedAvg. Four representative methods are shown for clarity.

## 5 Conclusion

In this paper, we demonstrated that selectively recycling updates in specific layers can reduce communication costs in FL while preserving model accuracy. In particular, our study empirically proved that the gradient-to-weight magnitude ratio can serve as a practical metric for identifying the least significant layers. This layer-wise partial model aggregation scheme is expected to facilitate the development of efficient FL applications and promote the partial model training paradigm across various deep learning fields. We consider developing a communication-efficient Large Language Model fine-tuning method based on the update recycling scheme as a promising direction for future work. A discussion of the broader impact of this work is provided in Appendix A.1.

## Acknowledgments

This work was partly supported by Institute of Information & communications Technology Planning & Evaluation (IITP) grant funded by the Korea government(MSIT) (No.RS-2022-00155915, Artificial Intelligence Convergence Innovation Human Resources Development (Inha University)) and National Research Foundation of Korea(NRF) grant funded by the Korea government(MSIT)(No. RS-2024-00452914).

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

# A   Appendix

The appendix is structured as follows:

- Section A.1 briefly announce broader impacts of this study.

- Section A.2 provides problem definition, assumptions, and proofs of our theoretical analysis.

- Section A.3 presents detailed experimental settings.

- Section A.4 presents additional experimental results and analyses.

## A.1   Broader Impacts

We do not anticipate any negative societal impact from our research. The proposed method accelerates neural network training in the context of Federated Learning. Faster training implies that target model accuracy can be achieved with fewer training iterations. As a result, it contributes to lower power consumption and a reduced carbon footprint.

## A.2   Theoretical Analysis

We consider non-convex and smooth optimization problems as follows.

$$\min_{x \in \mathbb{R}^d} F(x) := \frac{1}{m} \sum_{i=1}^{m} F_i(x), \tag{8}$$

where $F_i(x) = \mathbb{E}_{\xi_i \sim D_i}[f(x, \xi_i)]$ is the local loss function associated with the local data distribution $D_i$ of client $i$ and $m$ is the number of clients.

Our analysis is based on the following assumptions.

**Assumption 1.** *(Lipschitz continuity) There exists a constant $\mathcal{L} > 0$, such that $\|\nabla F_i(x) - \nabla F_i(y)\| \le \mathcal{L}\|x - y\|, \forall x, y \in \mathbb{R}^d,$ and $i \in [m]$.*

**Assumption 2.** *(Unbiased local gradients) The local gradient estimator is unbiased such that $\mathbb{E}_{\xi_i \sim D_i}[\nabla f(x, \xi_i)] = \nabla F_i(x), \forall i \in [m]$.*

**Assumption 3.** *(Bounded local and global variance) There exist two constants $\sigma_L > 0$ and $\sigma_G > 0$, such that the local gradient variance is bounded by $\mathbb{E}[\|\nabla f(x, \xi_i) - \nabla F_i(x)\|^2] \le \sigma_L^2, \forall i \in [m]$, and the global variability is bounded by $\mathbb{E}\left[\|\nabla F_i(x) - \nabla F(x)\|^2\right] \le \sigma_G^2, \forall i \in [m]$.*

Herein, we analyze the convergence properties of FedAvg as follows. First, the following Lemma is a slightly refined version of Lemma 3 in [30]. This Lemma is also used as Lemma 2 in [38].

**Lemma 3.** *(model discrepancy) For any step-size satisfying $\eta \le \frac{1}{2\sqrt{3}L\tau}$, we have the following result:*

$$\frac{1}{m} \sum_{i=0}^{m} \mathbb{E}[\|\mathbf{x}_{t,k}^i - \mathbf{x}_t\|^2] \le 5\eta^2 \sigma_L^2 + 30\tau\eta^2 \sigma_G^2 + 30\tau\eta^2 \mathbb{E}[\|\nabla F(\mathbf{x}_t)\|^2].$$

*Proof.* For any client $i \in [m]$, $t \in [T-1]$, and $k \in [\tau]$, we have

$$\mathbb{E}[\|\mathbf{x}_{t,k}^i - \mathbf{x}_t\|^2] = \mathbb{E}[\|\mathbf{x}_{t,k-1}^i - \mathbf{x}_t - \eta g_{t,k-1}^i\|^2]$$

$$= \mathbb{E}[\|\mathbf{x}_{t,k-1}^i - \mathbf{x}_t - \eta \left(g_{t,k-1}^i - \nabla F_i(\mathbf{x}_{t,k-1}^i) + \nabla F_i(\mathbf{x}_{t,k-1}^i) - \nabla F_i(\mathbf{x}_t) + \nabla F_i(\mathbf{x}_t) - \nabla F(\mathbf{x}_t) + \nabla F(\mathbf{x}_t)\right)\|^2]$$

$$= \mathbb{E}[\|\eta \left(g_{t,k-1}^i - \nabla F_i(\mathbf{x}_{t,k-1}^i)\right)\|^2]$$
$$\quad + 2\mathbb{E}[\langle \eta \left(g_{t,k-1}^i - \nabla F_i(\mathbf{x}_{t,k-1}^i)\right), \mathbf{x}_{t,k-1}^i - \mathbf{x}_t - \eta \left(\nabla F_i(\mathbf{x}_{t,k-1}^i) - \nabla F_i(\mathbf{x}_t) + \nabla F_i(\mathbf{x}_t) - \nabla F(\mathbf{x}_t) + \nabla F(\mathbf{x}_t)\right)\rangle]$$
$$\quad + \mathbb{E}[\|\mathbf{x}_{t,k-1}^i - \mathbf{x}_t - \eta \left(\nabla F_i(\mathbf{x}_{t,k-1}^i) - \nabla F_i(\mathbf{x}_t) + \nabla F_i(\mathbf{x}_t) - \nabla F(\mathbf{x}_t) + \nabla F(\mathbf{x}_t)\right)\|^2]$$

$$= \mathbb{E}[\|\eta \left(g_{t,k-1}^i - \nabla F_i(\mathbf{x}_{t,k-1}^i)\right)\|^2] \tag{9}$$
$$\quad + \mathbb{E}[\|\mathbf{x}_{t,k-1}^i - \mathbf{x}_t - \eta \left(\nabla F_i(\mathbf{x}_{t,k-1}^i) - \nabla F_i(\mathbf{x}_t) + \nabla F_i(\mathbf{x}_t) - \nabla F(\mathbf{x}_t) + \nabla F(\mathbf{x}_t)\right)\|^2]$$

$$\leq \eta^2 \sigma_L^2 + \mathbb{E}[\|\mathbf{x}_{t,k-1}^i - \mathbf{x}_t - \eta \left(\nabla F_i(\mathbf{x}_{t,k-1}^i) - \nabla F_i(\mathbf{x}_t) + \nabla F_i(\mathbf{x}_t) - \nabla F(\mathbf{x}_t) + \nabla F(\mathbf{x}_t)\right)\|^2]$$

$$\leq \eta^2 \sigma_L^2 + \left(1 + \frac{1}{2\tau - 1}\right) \mathbb{E}[\|\mathbf{x}_{t,k-1}^i - \mathbf{x}_t\|^2] \tag{10}$$
$$\quad + 2\tau\eta^2 \mathbb{E}[\|\nabla F_i(\mathbf{x}_{t,k-1}^i) - \nabla F_i(\mathbf{x}_t) + \nabla F_i(\mathbf{x}_t) - \nabla F(\mathbf{x}_t) + \nabla F(\mathbf{x}_t)\|^2]$$

$$\leq \eta^2 \sigma_L^2 + \left(1 + \frac{1}{2\tau - 1}\right) \mathbb{E}[\|\mathbf{x}_{t,k-1}^i - \mathbf{x}_t\|^2]$$
$$\quad + 6\tau\eta^2 \left(\mathbb{E}[\|\nabla F_i(\mathbf{x}_{t,k-1}^i) - \nabla F_i(\mathbf{x}_t)\|^2] + \mathbb{E}[\|\nabla F_i(\mathbf{x}_t) - \nabla F(\mathbf{x}_t)\|^2] + \mathbb{E}[\|\nabla F(\mathbf{x}_t)\|^2]\right)$$

$$\leq \eta^2 \sigma_L^2 + \left(1 + \frac{1}{2\tau - 1}\right) \mathbb{E}[\|\mathbf{x}_{t,k-1}^i - \mathbf{x}_t\|^2]$$
$$\quad + 6\tau\eta^2 \sigma_G^2 + 6\tau\eta^2 L^2 \mathbb{E}[\|\mathbf{x}_{t,k-1}^i - \mathbf{x}_t\|^2] + 6\tau\eta^2 \mathbb{E}[\|\nabla F(\mathbf{x}_t)\|^2]$$

$$= \eta^2 \sigma_L^2 + 6\tau\eta^2 \sigma_G^2 + \left(1 + \frac{1}{2\tau - 1} + 6\tau\eta^2 L^2\right) \mathbb{E}[\|\mathbf{x}_{t,k-1}^i - \mathbf{x}_t\|^2] + 6\tau\eta^2 \mathbb{E}[\|\nabla F(\mathbf{x}_t)\|^2],$$

where (9) is because $\mathbb{E}[g_{t,k-1}^i] = \nabla F_i(\mathbf{x}_{t,k}^i)$. The (10) is based on the fact that

$$\|\mathbf{a} + \mathbf{b}\|^2 \leq (1 + \frac{1}{\alpha})\|\mathbf{a}\|^2 + (1 + \alpha)\|\mathbf{b}\|^2$$

for any $\alpha > 0$.

Next, if $\eta \leq \frac{1}{2\sqrt{3}L\tau}$, the above bound can be simplified as follows.

$$\mathbb{E}[\|\mathbf{x}_{t,k}^i - \mathbf{x}_t\|^2]$$
$$\leq \eta^2 \sigma_L^2 + 6\tau\eta^2 \sigma_G^2 + \left(1 + \frac{1}{2\tau - 1} + 6\tau\eta^2 L^2\right) \mathbb{E}[\|\mathbf{x}_{t,k-1}^i - \mathbf{x}_t\|^2] + 6\tau\eta^2 \mathbb{E}[\|\nabla F(\mathbf{x}_t)\|^2]$$
$$\leq \eta^2 \sigma_L^2 + 6\tau\eta^2 \sigma_G^2 + \left(1 + \frac{1}{\tau - 1}\right) \mathbb{E}[\|\mathbf{x}_{t,k-1}^i - \mathbf{x}_t\|^2] + 6\tau\eta^2 \mathbb{E}[\|\nabla F(\mathbf{x}_t)\|^2].$$

Then, by unrolling the recursion until $k - 1$ goes to 0, we have

$$\mathbb{E}[\|\mathbf{x}_{t,k}^i - \mathbf{x}_t\|^2] \leq \sum_{j=0}^{k-1} \left(1 + \frac{1}{\tau - 1}\right)^j \left(\eta^2 \sigma_L^2 + 6\tau\eta^2 \sigma_G^2 + 6\tau\eta^2 \mathbb{E}[\|\nabla F(\mathbf{x}_t)\|^2]\right)$$

$$\leq (\tau - 1)((1 + \frac{1}{\tau - 1})^{k-1} - 1) \left(\eta^2 \sigma_L^2 + 6\tau\eta^2 \sigma_G^2 + 6\tau\eta^2 \mathbb{E}[\|\nabla F(\mathbf{x}_t)\|^2]\right)$$

$$\leq (\tau - 1)((1 + \frac{1}{\tau - 1})^\tau - 1) \left(\eta^2 \sigma_L^2 + 6\tau\eta^2 \sigma_G^2 + 6\tau\eta^2 \mathbb{E}[\|\nabla F(\mathbf{x}_t)\|^2]\right)$$

$$\leq 5\tau\eta^2 \sigma_L^2 + 30\tau^2\eta^2 \sigma_G^2 + 30\tau^2\eta^2 \mathbb{E}[\|\nabla F(\mathbf{x}_t)\|^2]. \tag{11}$$

where (11) is because that the maximum value of $(\tau - 1)((1 + \frac{1}{\tau-1})^\tau - 1)$ is $\frac{19}{4}$ when $\tau = 3$. Finally, because the right-hand side of (11) is independent of $m$, we have

$$\frac{1}{m} \sum_{i=0}^m \mathbb{E}[\|\mathbf{x}_{t,k}^i - \mathbf{x}_t\|^2] \leq 5\tau\eta^2 \sigma_L^2 + 30\tau^2\eta^2 \sigma_G^2 + 30\tau^2\eta^2 \mathbb{E}[\|\nabla F(\mathbf{x}_t)\|^2].$$

$\square$

Based on the proposed update recycling method, the noise $n_t$ is defined as follows.

$$n_t = \frac{1}{m} \sum_{i=1}^{m} \sum_{j=0}^{\tau-1} \left( \hat{g}_{t-k,j}^i - \hat{g}_{t,j}^i \right),$$

where $\hat{g}$ indicates the gradient vector that has non-zero gradients only at the layers where their updates will be recycled.

**Lemma 4.** *(noise) Under assumption $1 \sim 3$, if the learning rate $\eta \le \frac{1}{\mathcal{L}\tau}$, the accumulated noise is bounded as follows.*

$$\sum_{t=0}^{T-1} \mathbb{E}\left[ \|n_t\|^2 \right] \le 4T\tau^2 \sigma_L^2 + 8T\tau^2 \sigma_G^2 + 8\kappa\tau^2 \sum_{t=0}^{T-1} \mathbb{E}\left[ \|\nabla F(\mathbf{x}_t)\|^2 \right]$$

$$+ \frac{8\tau L^2}{m} \sum_{t=0}^{T-1} \sum_{i=1}^{m} \sum_{j=0}^{\tau-1} \mathbb{E}\left[ \|\mathbf{x}_{t,j}^i - \mathbf{x}_t\|^2 \right], \tag{12}$$

*where $\kappa$ is the ratio of $\|\nabla \hat{F}(\mathbf{x}_t)\|^2$ to $\|\nabla F(\mathbf{x}_t)\|^2$.*

*Proof.*

$$\mathbb{E}\left[ \|n_t\|^2 \right] = \mathbb{E}\left[ \left\| \frac{1}{m} \sum_{i=1}^{m} \sum_{j=0}^{\tau-1} \left( \hat{g}_{t-k,j}^i - \hat{g}_{t,j}^i \right) \right\|^2 \right]$$

$$\le 2\mathbb{E}\left[ \left\| \frac{1}{m} \sum_{i=1}^{m} \sum_{j=0}^{\tau-1} \hat{g}_{t-k,j}^i \right\|^2 \right] + 2\mathbb{E}\left[ \left\| \frac{1}{m} \sum_{i=1}^{m} \sum_{j=0}^{\tau-1} \hat{g}_{t,j}^i \right\|^2 \right]$$

$$\le \frac{2\tau}{m} \sum_{i=1}^{m} \sum_{j=0}^{\tau-1} \mathbb{E}\left[ \|\hat{g}_{t-k,j}^i\|^2 \right] + \frac{2\tau}{m} \sum_{i=1}^{m} \sum_{j=0}^{\tau-1} \mathbb{E}\left[ \|\hat{g}_{t,j}^i\|^2 \right]$$

$$= \frac{2\tau}{m} \sum_{i=1}^{m} \sum_{j=0}^{\tau-1} \mathbb{E}\left[ \left\| \hat{g}_{t-k,j}^i - \nabla \hat{F}_i(\mathbf{x}_{t-k,j}^i) + \nabla \hat{F}_i(\mathbf{x}_{t-k,j}^i) \right\|^2 \right]$$

$$+ \frac{2\tau}{m} \sum_{i=1}^{m} \sum_{j=0}^{\tau-1} \mathbb{E}\left[ \left\| \hat{g}_{t,j}^i - \nabla \hat{F}_i(\mathbf{x}_{t,j}^i) + \nabla \hat{F}_i(\mathbf{x}_{t,j}^i) \right\|^2 \right]$$

$$= \frac{2\tau}{m} \sum_{i=1}^{m} \sum_{j=0}^{\tau-1} \left( \mathbb{E}\left[ \left\| \hat{g}_{t-k,j}^i - \nabla \hat{F}_i(\mathbf{x}_{t-k,j}^i) \right\|^2 \right] + \mathbb{E}\left[ \left\| \nabla \hat{F}_i(\mathbf{x}_{t-k,j}^i) \right\|^2 \right] \right) \tag{13}$$

$$+ \frac{2\tau}{m} \sum_{i=1}^{m} \sum_{j=0}^{\tau-1} \left( \mathbb{E}\left[ \left\| \hat{g}_{t,j}^i - \nabla \hat{F}_i(\mathbf{x}_{t,j}^i) \right\|^2 \right] + \mathbb{E}\left[ \left\| \nabla \hat{F}_i(\mathbf{x}_{t,j}^i) \right\|^2 \right] \right)$$

$$\le 2\tau^2 \sigma_L^2 + \frac{2\tau}{m} \sum_{i=1}^{m} \sum_{j=0}^{\tau-1} \mathbb{E}\left[ \left\| \nabla \hat{F}_i(\mathbf{x}_{t-k,j}^i) \right\|^2 \right] + 2\tau^2 \sigma_L^2 + \frac{2\tau}{m} \sum_{i=1}^{m} \sum_{j=0}^{\tau-1} \mathbb{E}\left[ \left\| \nabla \hat{F}_i(\mathbf{x}_{t,j}^i) \right\|^2 \right].$$

where (13) is based on the fact that $\mathbb{E}[\|\mathbf{x}\|^2] = \mathbb{E}[\|\mathbf{x} - \mathbb{E}[\mathbf{x}]\|^2] + \|\mathbb{E}[\mathbf{x}]\|^2$. Then, the right-hand side can be further bounded as follows.

$$
\begin{aligned}
\mathbb{E}\left[\|n_t\|^2\right] &\leq 4\tau^2\sigma_L^2 + \frac{2\tau}{m}\sum_{i=1}^{m}\sum_{j=0}^{\tau-1}\mathbb{E}\left[\left\|\nabla\hat{F}_i(\mathbf{x}_{t-k,j}^i)\right\|^2\right] + \frac{2\tau}{m}\sum_{i=1}^{m}\sum_{j=0}^{\tau-1}\mathbb{E}\left[\left\|\nabla\hat{F}_i(\mathbf{x}_{t,j}^i)\right\|^2\right] \\
&= 4\tau^2\sigma_L^2 + \frac{2\tau}{m}\sum_{i=1}^{m}\sum_{j=0}^{\tau-1}\mathbb{E}\left[\left\|\nabla\hat{F}_i(\mathbf{x}_{t-k,j}^i) - \nabla\hat{F}_i(\mathbf{x}_{t-k}) + \nabla\hat{F}_i(\mathbf{x}_{t-k})\right\|^2\right] \\
&\quad + \frac{2\tau}{m}\sum_{i=1}^{m}\sum_{j=0}^{\tau-1}\mathbb{E}\left[\left\|\nabla\hat{F}_i(\mathbf{x}_{t,j}^i) - \nabla\hat{F}_i(\mathbf{x}_t) + \nabla\hat{F}_i(\mathbf{x}_t)\right\|^2\right] \\
&\leq 4\tau^2\sigma_L^2 + \frac{4\tau}{m}\sum_{i=1}^{m}\sum_{j=0}^{\tau-1}\mathbb{E}\left[\left\|\nabla\hat{F}_i(\mathbf{x}_{t-k,j}^i) - \nabla\hat{F}_i(\mathbf{x}_{t-k})\right\|^2\right] + \frac{4\tau}{m}\sum_{i=1}^{m}\sum_{j=0}^{\tau-1}\mathbb{E}\left[\left\|\nabla\hat{F}_i(\mathbf{x}_{t-k})\right\|^2\right] \\
&\quad + \frac{4\tau}{m}\sum_{i=1}^{m}\sum_{j=0}^{\tau-1}\mathbb{E}\left[\left\|\nabla\hat{F}_i(\mathbf{x}_{t,j}^i) - \nabla\hat{F}_i(\mathbf{x}_t)\right\|^2\right] + \frac{4\tau}{m}\sum_{i=1}^{m}\sum_{j=0}^{\tau-1}\mathbb{E}\left[\left\|\nabla\hat{F}_i(\mathbf{x}_t)\right\|^2\right] \\
&\leq 4\tau^2\sigma_L^2 + \frac{4\tau L^2}{m}\sum_{i=1}^{m}\sum_{j=0}^{\tau-1}\mathbb{E}\left[\left\|\mathbf{x}_{t-k,j}^i - \mathbf{x}_{t-k}\right\|^2\right] + \frac{4\tau}{m}\sum_{i=1}^{m}\sum_{j=0}^{\tau-1}\mathbb{E}\left[\left\|\nabla\hat{F}_i(\mathbf{x}_{t-k})\right\|^2\right] \\
&\quad + \frac{4\tau L^2}{m}\sum_{i=1}^{m}\sum_{j=0}^{\tau-1}\mathbb{E}\left[\left\|\mathbf{x}_{t,j}^i - \mathbf{x}_t\right\|^2\right] + \frac{4\tau}{m}\sum_{i=1}^{m}\sum_{j=0}^{\tau-1}\mathbb{E}\left[\left\|\nabla\hat{F}_i(\mathbf{x}_t)\right\|^2\right] \\
&= 4\tau^2\sigma_L^2 + \frac{4\tau L^2}{m}\sum_{i=1}^{m}\sum_{j=0}^{\tau-1}\mathbb{E}\left[\left\|\mathbf{x}_{t-k,j}^i - \mathbf{x}_{t-k}\right\|^2\right] + \frac{4\tau L^2}{m}\sum_{i=1}^{m}\sum_{j=0}^{\tau-1}\mathbb{E}\left[\left\|\mathbf{x}_{t,j}^i - \mathbf{x}_t\right\|^2\right] \\
&\quad + \frac{4\tau}{m}\sum_{i=1}^{m}\sum_{j=0}^{\tau-1}\mathbb{E}\left[\left\|\nabla\hat{F}_i(\mathbf{x}_{t-k}) - \nabla\hat{F}(\mathbf{x}_{t-k}) + \nabla\hat{F}(\mathbf{x}_{t-k})\right\|^2\right] \\
&\quad + \frac{4\tau}{m}\sum_{i=1}^{m}\sum_{j=0}^{\tau-1}\mathbb{E}\left[\left\|\nabla\hat{F}_i(\mathbf{x}_t) - \nabla\hat{F}(\mathbf{x}_t) + \nabla\hat{F}(\mathbf{x}_t)\right\|^2\right] \\
&\leq 4\tau^2\sigma_L^2 + \frac{4\tau L^2}{m}\sum_{i=1}^{m}\sum_{j=0}^{\tau-1}\mathbb{E}\left[\left\|\mathbf{x}_{t-k,j}^i - \mathbf{x}_{t-k}\right\|^2\right] + \frac{4\tau L^2}{m}\sum_{i=1}^{m}\sum_{j=0}^{\tau-1}\mathbb{E}\left[\left\|\mathbf{x}_{t,j}^i - \mathbf{x}_t\right\|^2\right] \\
&\quad + \frac{4\tau}{m}\sum_{i=1}^{m}\sum_{j=0}^{\tau-1}\mathbb{E}\left[\left\|\nabla\hat{F}_i(\mathbf{x}_{t-k}) - \nabla\hat{F}(\mathbf{x}_{t-k})\right\|^2\right] + \frac{4\tau}{m}\sum_{i=1}^{m}\sum_{j=0}^{\tau-1}\mathbb{E}\left[\left\|\nabla\hat{F}(\mathbf{x}_{t-k})\right\|^2\right] \\
&\quad + \frac{4\tau}{m}\sum_{i=1}^{m}\sum_{j=0}^{\tau-1}\mathbb{E}\left[\left\|\nabla\hat{F}_i(\mathbf{x}_t) - \nabla\hat{F}(\mathbf{x}_t)\right\|^2\right] + \frac{4\tau}{m}\sum_{i=1}^{m}\sum_{j=0}^{\tau-1}\mathbb{E}\left[\left\|\nabla\hat{F}(\mathbf{x}_t)\right\|^2\right] \\
&= 4\tau^2\sigma_L^2 + 8\tau^2\sigma_G^2 + 4\tau^2\mathbb{E}\left[\left\|\nabla\hat{F}(\mathbf{x}_{t-k})\right\|^2\right] + 4\tau^2\mathbb{E}\left[\left\|\nabla\hat{F}(\mathbf{x}_t)\right\|^2\right] \qquad (14) \\
&\quad + \frac{4\tau L^2}{m}\sum_{i=1}^{m}\sum_{j=0}^{\tau-1}\mathbb{E}\left[\left\|\mathbf{x}_{t-k,j}^i - \mathbf{x}_{t-k}\right\|^2\right] + \frac{4\tau L^2}{m}\sum_{i=1}^{m}\sum_{j=0}^{\tau-1}\mathbb{E}\left[\left\|\mathbf{x}_{t,j}^i - \mathbf{x}_t\right\|^2\right],
\end{aligned}
$$

where (14) follows $\|\nabla \hat{F}(\cdot)\|^2 \le \|\nabla F(\cdot)\|^2$. By summing up $\mathbb{E}[\|n_t\|^2]$ across $T$ rounds, we have

$$
\begin{aligned}
\sum_{t=0}^{T-1} \mathbb{E}\left[\|n_t\|^2\right] &\le 4T\tau^2\sigma_L^2 + 8T\tau^2\sigma_G^2 + 4\tau^2 \sum_{t=0}^{T-1} \mathbb{E}\left[\left\|\nabla\hat{F}(\mathbf{x}_{t-k})\right\|^2\right] + 4\tau^2 \sum_{t=0}^{T-1} \mathbb{E}\left[\left\|\nabla\hat{F}(\mathbf{x}_t)\right\|^2\right] \\
&\quad + \frac{4\tau L^2}{m} \sum_{t=0}^{T-1}\sum_{i=1}^{m}\sum_{j=0}^{\tau-1} \mathbb{E}\left[\left\|\mathbf{x}_{t-k,j}^i - \mathbf{x}_{t-k}\right\|^2\right] + \frac{4\tau L^2}{m} \sum_{t=0}^{T-1}\sum_{i=1}^{m}\sum_{j=0}^{\tau-1} \mathbb{E}\left[\left\|\mathbf{x}_{t,j}^i - \mathbf{x}_t\right\|^2\right] \\
&\le 4T\tau^2\sigma_L^2 + 8T\tau^2\sigma_G^2 + 8\tau^2 \sum_{t=0}^{T-1} \mathbb{E}\left[\left\|\nabla\hat{F}(\mathbf{x}_t)\right\|^2\right] + \frac{8\tau L^2}{m} \sum_{t=0}^{T-1}\sum_{i=1}^{m}\sum_{j=0}^{\tau-1} \mathbb{E}\left[\left\|\mathbf{x}_{t,j}^i - \mathbf{x}_t\right\|^2\right] \\
&\le 4T\tau^2\sigma_L^2 + 8T\tau^2\sigma_G^2 + 8\kappa\tau^2 \sum_{t=0}^{T-1} \mathbb{E}\left[\|\nabla F(\mathbf{x}_t)\|^2\right] + \frac{8\tau L^2}{m} \sum_{t=0}^{T-1}\sum_{i=1}^{m}\sum_{j=0}^{\tau-1} \mathbb{E}\left[\left\|\mathbf{x}_{t,j}^i - \mathbf{x}_t\right\|^2\right],
\end{aligned}
$$

where $\kappa$ is the ratio of the recycled update norm to the full update norm. Because all the gradients at the layers not recycled are zeroed out, the ratio $\kappa$ lies between 0 and 1; $0 < \kappa < 1$. $\qquad\square$

**Lemma 5.** *(framework) Under assumption 1 $\sim$ 3, if the learning rate $\eta \le \frac{1}{\mathcal{L}\tau}$, we have*

$$
\begin{aligned}
\sum_{t=0}^{T-1} \mathbb{E}\left[\|\nabla F(\mathbf{x}_t)\|^2\right] &\le \frac{2}{(1-16\kappa)\eta\tau}\left(F(\mathbf{x}_0) - F(\mathbf{x}_T)\right) + \frac{2T}{1-16\kappa}\left(\frac{\mathcal{L}\eta}{m} + 4\right)\sigma_L^2 + \frac{16T}{1-16\kappa}\sigma_G^2 \\
&\quad + \frac{18\mathcal{L}^2}{(1-16\kappa)m\tau} \sum_{t=0}^{T-1}\sum_{i=1}^{m}\sum_{j=0}^{\tau-1} \mathbb{E}\left[\left\|\mathbf{x}_{t,j}^i - \mathbf{x}_t\right\|^2\right],
\end{aligned}
$$

*where $\kappa$ is the ratio of the norm of the recycling layers' gradients: $\|\nabla\hat{F}(\mathbf{x}_t)\|^2$ to that of the full model gradients: $\|\nabla F(\mathbf{x}_t)\|^2$.*

*Proof.* We first define the following notations for convenience.

$$
\begin{aligned}
\Delta_t^i &= \sum_{j=0}^{\tau-1} g_{t,j}^i := \sum_{j=0}^{\tau-1} \nabla f(\mathbf{x}_{t,j}^i, \xi_j^i) \\
\Delta_t &:= \frac{1}{m}\sum_{i=1}^{m} \Delta_t^i + n_t,
\end{aligned}
$$

where $\xi_j^i$ is a random sample drawn from the local dataset $i$ at the local step $j$ and $n_t$ is a noise caused by the update recycling.

Based on Assumption 1, taking expectation of $F(\mathbf{x}_{t+1})$, we have:

$$
\begin{aligned}
\mathbb{E}[F(\mathbf{x}_{t+1})] &\le F(\mathbf{x}_t) + \langle\nabla F(\mathbf{x}_t), \mathbb{E}[\mathbf{x}_{t+1} - \mathbf{x}_t]\rangle + \frac{\mathcal{L}}{2}\mathbb{E}[\|\mathbf{x}_{t+1} - \mathbf{x}_t\|^2] \\
&= F(\mathbf{x}_t) + \langle\nabla F(\mathbf{x}_t), \mathbb{E}[-\eta\Delta_t]\rangle + \frac{\mathcal{L}}{2}\mathbb{E}[\|\eta\Delta_t\|^2] \\
&= F(\mathbf{x}_t) + \langle\nabla F(\mathbf{x}_t), \mathbb{E}[-\eta\Delta_t + \eta\tau\nabla F(\mathbf{x}_t) - \eta\tau\nabla F(\mathbf{x}_t)]\rangle + \frac{\mathcal{L}}{2}\mathbb{E}[\|\eta\Delta_t\|^2] \\
&= F(\mathbf{x}_t) - \eta\tau\|\nabla F(\mathbf{x}_t)\|^2 + \underbrace{\langle\nabla F(\mathbf{x}_t), \mathbb{E}[-\eta\Delta_t + \eta\tau\nabla F(\mathbf{x}_t)]\rangle}_{T_1} + \frac{\mathcal{L}}{2}\underbrace{\mathbb{E}[\|\eta\Delta_t\|^2]}_{T_2}.
\end{aligned}
$$

$$(15)$$

Now, let us bound $T_1$ and $T_2$ separately as follows.

**Bounding $T_1$.**

$$T_1 = \langle \nabla F(\mathbf{x}_t), \mathbb{E}\left[-\eta \Delta_t + \eta\tau \nabla F(\mathbf{x}_t)\right]\rangle$$

$$= \left\langle \nabla F(\mathbf{x}_t), \mathbb{E}\left[-\eta\left(\frac{1}{m}\sum_{i=0}^{m}\sum_{j=0}^{\tau-1} g_{t,j}^i + n_t\right) + \eta\tau\nabla F(\mathbf{x}_t)\right]\right\rangle$$

$$= \left\langle \nabla F(\mathbf{x}_t), \mathbb{E}\left[-\frac{\eta}{m}\sum_{i=0}^{m}\sum_{j=0}^{\tau-1} g_{t,j}^i + \eta\tau\nabla F(\mathbf{x}_t) - \eta n_t\right]\right\rangle$$

$$= \left\langle \nabla F(\mathbf{x}_t), \mathbb{E}\left[-\frac{\eta}{m}\sum_{i=0}^{m}\sum_{j=0}^{\tau-1} \nabla F_i(\mathbf{x}_{t,j}^i) + \frac{\eta}{m}\sum_{i=1}^{m}\sum_{j=0}^{\tau-1} \nabla F_i(\mathbf{x}_t) - \eta n_t\right]\right\rangle$$

$$= \left\langle \nabla F(\mathbf{x}_t), \mathbb{E}\left[-\frac{\eta}{m}\sum_{i=0}^{m}\sum_{j=0}^{\tau-1} \left(\nabla F_i(\mathbf{x}_{t,j}^i) - \nabla F_i(\mathbf{x}_t)\right) - \eta n_t\right]\right\rangle$$

$$= \left\langle \sqrt{\eta\tau}\nabla F(\mathbf{x}_t), \mathbb{E}\left[-\frac{\sqrt{\eta}}{m\sqrt{\tau}}\sum_{i=0}^{m}\sum_{j=0}^{\tau-1} \left(\nabla F_i(\mathbf{x}_{t,j}^i) - \nabla F_i(\mathbf{x}_t)\right) - \frac{\sqrt{\eta}}{\sqrt{\tau}}n_t\right]\right\rangle$$

$$= \frac{\eta\tau}{2}\|\nabla F(\mathbf{x}_t)\|^2 + \frac{1}{2}\mathbb{E}\left[\left\|\frac{\sqrt{\eta}}{m\sqrt{\tau}}\sum_{i=1}^{m}\sum_{j=0}^{\tau-1}\left(\nabla F_i(\mathbf{x}_{t,j}^i) - \nabla F_i(\mathbf{x}_t)\right) + \frac{\sqrt{\eta}}{\sqrt{\tau}}n_t\right\|^2\right]$$

$$- \frac{1}{2}\mathbb{E}\left[\left\|\frac{\sqrt{\eta}}{m\sqrt{\tau}}\sum_{i=1}^{m}\sum_{j=0}^{\tau-1}\nabla F_i(\mathbf{x}_{t,j}^i) + \frac{\sqrt{\eta}}{\sqrt{\tau}}n_t\right\|^2\right] \tag{16}$$

$$\leq \frac{\eta\tau}{2}\|\nabla F(\mathbf{x}_t)\|^2 + \mathbb{E}\left[\left\|\frac{\sqrt{\eta}}{m\sqrt{\tau}}\sum_{i=1}^{m}\sum_{j=0}^{\tau-1}\left(\nabla F_i(\mathbf{x}_{t,j}^i) - \nabla F_i(\mathbf{x}_t)\right)\right\|^2\right] + \mathbb{E}\left[\left\|\frac{\sqrt{\eta}}{\sqrt{\tau}}n_t\right\|^2\right]$$

$$- \frac{\eta}{2\tau}\mathbb{E}\left[\left\|\frac{1}{m}\sum_{i=1}^{m}\sum_{j=0}^{\tau-1}\nabla F_i(\mathbf{x}_{t,j}^i) + n_t\right\|^2\right]$$

$$\leq \frac{\eta\tau}{2}\|\nabla F(\mathbf{x}_t)\|^2 + \frac{\eta}{m}\sum_{i=1}^{m}\sum_{j=0}^{\tau-1}\mathbb{E}\left[\left\|\nabla F_i(\mathbf{x}_{t,j}^i) - \nabla F_i(\mathbf{x}_t)\right\|^2\right] + \frac{\eta}{\tau}\mathbb{E}\left[\|n_t\|^2\right]$$

$$- \frac{\eta}{2\tau}\mathbb{E}\left[\left\|\frac{1}{m}\sum_{i=1}^{m}\sum_{j=0}^{\tau-1}\nabla F_i(\mathbf{x}_{t,j}^i) + n_t\right\|^2\right] \tag{17}$$

$$\leq \frac{\eta\tau}{2}\|\nabla F(\mathbf{x}_t)\|^2 + \frac{\eta\mathcal{L}^2}{m}\sum_{i=1}^{m}\sum_{j=0}^{\tau-1}\mathbb{E}\left[\left\|\mathbf{x}_{t,j}^i - \mathbf{x}_t\right\|^2\right] + \frac{\eta}{\tau}\mathbb{E}\left[\|n_t\|^2\right]$$

$$- \frac{\eta}{2m^2\tau}\mathbb{E}\left[\left\|\sum_{i=1}^{m}\sum_{j=0}^{\tau-1}\left(\nabla F_i(\mathbf{x}_{t,j}^i) + \frac{1}{\tau}n_t\right)\right\|^2\right], \tag{18}$$

where (16) holds because $\langle x, y \rangle = \frac{1}{2}[\|x\|^2 + \|y\|^2 - \|x-y\|^2]$ for $x = \sqrt{\eta\tau}\nabla F(\mathbf{x}_t)$ and $\mathbf{y} = -\frac{\sqrt{\eta}}{m\sqrt{\tau}}\sum_{i=1}^{m}\sum_{j=0}^{\tau-1}(\nabla F_i(\mathbf{x}_{t,j}^i) - \nabla F_i(\mathbf{x}_t))$. Also, (17) is based on the convexity of $\ell_2$ norm and Jensen's inequality.

**Bounding $T_2$.**

$$T_2 = \mathbb{E}[\|\eta\Delta_t\|^2] = \eta^2 \mathbb{E}\left[\left\|\frac{1}{m}\sum_{i=1}^{m}\sum_{j=0}^{\tau}g_{t,j}^i + n_t\right\|^2\right]$$

$$= \eta^2 \mathbb{E}\left[\left\|\frac{1}{m}\sum_{i=1}^{m}\sum_{j=0}^{\tau}\left(g_{t,j}^i + \frac{1}{\tau}n_t - \nabla F_i(\mathbf{x}_{t,j}^i) + \nabla F_i(\mathbf{x}_{t,j}^i)\right)\right\|^2\right]$$

$$\leq 2\eta^2 \mathbb{E}\left[\left\|\frac{1}{m}\sum_{i=1}^{m}\sum_{j=0}^{\tau}\left(g_{t,j}^i - \nabla F_i(\mathbf{x}_{t,j}^i)\right)\right\|^2\right] + 2\eta^2 \mathbb{E}\left[\left\|\frac{1}{m}\sum_{i=1}^{m}\sum_{j=0}^{\tau}\left(\nabla F_i(\mathbf{x}_{t,j}^i) + \frac{1}{\tau}n_t\right)\right\|^2\right]$$

$$= \frac{2\eta^2}{m^2}\sum_{i=1}^{m}\mathbb{E}\left[\left\|\sum_{j=0}^{\tau}g_{t,j}^i - \nabla F_i(\mathbf{x}_{t,j}^i)\right\|^2\right] + 2\eta^2 \mathbb{E}\left[\left\|\frac{1}{m}\sum_{i=1}^{m}\sum_{j=0}^{\tau}\nabla F_i(\mathbf{x}_{t,j}^i) + n_t\right\|^2\right] \tag{19}$$

$$\leq \frac{2\eta^2\tau}{m}\sigma_L^2 + \frac{2\eta^2}{m^2}\mathbb{E}\left[\left\|\sum_{i=1}^{m}\sum_{j=0}^{\tau}\left(\nabla F_i(\mathbf{x}_{t,j}^i) + \frac{1}{\tau}n_t\right)\right\|^2\right], \tag{20}$$

where (19) is due to the fact that $\mathbb{E}[\|\mathbf{x}_1 + \mathbf{x}_2 + \cdots + \mathbf{x}_n\|^2] = \mathbb{E}[\|\mathbf{x}_1\|^2 + \|\mathbf{x}_2\|^2 + \cdots + \|\mathbf{x}_n\|^2]$ if $\mathbf{x}_i$s are independent of each other with zero mean and $\mathbb{E}[g_{t,j}^i] = \nabla F_i(\mathbf{x}_{t,j}^i)$.

Now, by plugging in (18) and (20) into (15), we have

$$\mathbb{E}[F(\mathbf{x}_{t+1})] \leq F(\mathbf{x}_t) - \frac{\eta\tau}{2}\mathbb{E}\left[\|\nabla F(\mathbf{x}_t)\|^2\right] + \frac{\mathcal{L}\eta^2\tau}{m}\sigma_L^2 + \frac{\eta\mathcal{L}^2}{m}\sum_{i=1}^{m}\sum_{j=0}^{\tau-1}\mathbb{E}\left[\|\mathbf{x}_{t,j}^i - \mathbf{x}_t\|^2\right]$$

$$+ \left(\frac{\mathcal{L}\eta^2}{m^2} - \frac{\eta}{2m^2\tau}\right)\mathbb{E}\left[\left\|\sum_{i=1}^{m}\sum_{j=0}^{\tau}\left(\nabla F_i(\mathbf{x}_{t,j}^i) + \frac{1}{\tau}n_t\right)\right\|^2\right] + \frac{\eta}{\tau}\mathbb{E}\left[\|n_t\|^2\right]$$

$$\leq F(\mathbf{x}_t) - \frac{\eta\tau}{2}\mathbb{E}\left[\|\nabla F(\mathbf{x}_t)\|^2\right] + \frac{\mathcal{L}\eta^2\tau}{m}\sigma_L^2 + \frac{\eta}{\tau}\mathbb{E}\left[\|n_t\|^2\right]$$

$$+ \frac{\eta\mathcal{L}^2}{m}\sum_{i=1}^{m}\sum_{j=0}^{\tau-1}\mathbb{E}\left[\|\mathbf{x}_{t,j}^i - \mathbf{x}_t\|^2\right], \tag{21}$$

where (21) holds if $\eta \leq \frac{1}{\mathcal{L}\tau}$. Summing up (21) across $T$ communication rounds, we have

$$\sum_{t=0}^{T-1}(F(\mathbf{x}_{t+1}) - F(\mathbf{x}_t)) \leq -\frac{\eta\tau}{2}\sum_{t=0}^{T-1}\mathbb{E}\left[\|\nabla F(\mathbf{x}_t)\|^2\right] + \frac{\mathcal{L}\eta^2\tau T}{m}\sigma_L^2 + \frac{\eta}{\tau}\sum_{t=0}^{T-1}\mathbb{E}\left[\|n_t\|^2\right]$$

$$+ \frac{\eta\mathcal{L}^2}{m}\sum_{t=0}^{T-1}\sum_{i=1}^{m}\sum_{j=0}^{\tau-1}\mathbb{E}\left[\|\mathbf{x}_{t,j}^i - \mathbf{x}_t\|^2\right].$$

Based on Lemma 1, the right-hand side can be re-written as follows.

$$\sum_{t=0}^{T-1} (F(\mathbf{x}_{t+1}) - F(\mathbf{x}_t)) \leq \left(8\kappa\tau\eta - \frac{\eta\tau}{2}\right) \sum_{t=0}^{T-1} \mathbb{E}\left[\|\nabla F(\mathbf{x}_t)\|^2\right] + \frac{\mathcal{L}\eta^2\tau T}{m}\sigma_L^2 + 4\eta\tau T\sigma_L^2 + 8\eta\tau T\sigma_G^2$$

$$+ \left(\frac{8\eta\mathcal{L}^2}{m} + \frac{\eta\mathcal{L}^2}{m}\right) \sum_{t=0}^{T-1}\sum_{i=1}^{m}\sum_{j=0}^{\tau-1} \mathbb{E}\left[\left\|\mathbf{x}_{t,j}^i - \mathbf{x}_t\right\|^2\right]$$

$$= \left(8\kappa\tau\eta - \frac{\eta\tau}{2}\right) \sum_{t=0}^{T-1} \mathbb{E}\left[\|\nabla F(\mathbf{x}_t)\|^2\right] + \eta\tau T\left(\frac{\mathcal{L}\eta}{m} + 4\right)\sigma_L^2 + 8\eta\tau T\sigma_G^2$$

$$+ \frac{9\eta\mathcal{L}^2}{m} \sum_{t=0}^{T-1}\sum_{i=1}^{m}\sum_{j=0}^{\tau-1} \mathbb{E}\left[\left\|\mathbf{x}_{t,j}^i - \mathbf{x}_t\right\|^2\right]$$

After rearranging the telescoping sum, we finally have

$$\sum_{t=0}^{T-1} \mathbb{E}\left[\|\nabla F(\mathbf{x}_t)\|^2\right] \leq \frac{2}{(1-16\kappa)\eta\tau}(F(\mathbf{x}_0) - F(\mathbf{x}_T)) + \frac{2T}{1-16\kappa}\left(\frac{\mathcal{L}\eta}{m} + 4\right)\sigma_L^2 + \frac{16T}{1-16\kappa}\sigma_G^2$$

$$+ \frac{18\mathcal{L}^2}{(1-16\kappa)m\tau} \sum_{t=0}^{T-1}\sum_{i=1}^{m}\sum_{j=0}^{\tau-1} \mathbb{E}\left[\left\|\mathbf{x}_{t,j}^i - \mathbf{x}_t\right\|^2\right].$$

$$\square$$

Now, we can derive the following Theorem based on Lemma 3, 5, and 1 as follows.

**Theorem 6.** *Under assumption 1~3, if the learning rate $\eta \leq \frac{1-16\kappa}{6\sqrt{30}\mathcal{L}\tau}$ and $\kappa < \frac{1}{16}$, we have*

$$\sum_{t=0}^{T-1} \mathbb{E}\left[\|\nabla F(\mathbf{x}_t)\|^2\right] \leq \frac{4}{(1-16\kappa)\eta\tau}(F(\mathbf{x}_0) - F(\mathbf{x}_T)) + \frac{4T}{1-16\kappa}\left(\frac{\mathcal{L}\eta}{m} + 4 + 9\mathcal{L}^2\right)\sigma_L^2 + \frac{1080T\mathcal{L}^2\eta^2\tau^2}{1-16\kappa}\sigma_G^2.$$

*Proof.* Based on Lemma 3 and 5, we have

$$\sum_{t=0}^{T-1} \mathbb{E}\left[\|\nabla F(\mathbf{x}_t)\|^2\right] \leq \frac{2}{(1-16\kappa)\eta\tau}(F(\mathbf{x}_0) - F(\mathbf{x}_T)) + \frac{2T}{1-16\kappa}\left(\frac{\mathcal{L}\eta}{m} + 4\right)\sigma_L^2 + \frac{16T}{1-16\kappa}\sigma_G^2$$

$$+ \frac{18\mathcal{L}^2}{(1-16\kappa)m\tau} \sum_{t=0}^{T-1}\sum_{i=1}^{m}\sum_{j=0}^{\tau-1} \mathbb{E}\left[\left\|\mathbf{x}_{t,j}^i - \mathbf{x}_t\right\|^2\right]$$

$$\sum_{t=0}^{T-1} \mathbb{E}\left[\|\nabla F(\mathbf{x}_t)\|^2\right] \leq \frac{2}{(1-16\kappa)\eta\tau}(F(\mathbf{x}_0) - F(\mathbf{x}_T)) + \frac{2T}{1-16\kappa}\left(\frac{\mathcal{L}\eta}{m} + 4\right)\sigma_L^2 + \frac{16T}{1-16\kappa}\sigma_G^2$$

$$+ \frac{18\mathcal{L}^2}{(1-16\kappa)\tau} \sum_{t=0}^{T-1}\sum_{j=0}^{\tau-1} \left(5\eta^2\tau\sigma_L^2 + 30\eta^2\tau^2\sigma_G^2 + 30\eta^2\tau^2\mathbb{E}\left[\|\nabla F(\mathbf{x}_t)\|^2\right]\right)$$

$$\leq \frac{2}{(1-16\kappa)\eta\tau}(F(\mathbf{x}_0) - F(\mathbf{x}_T)) + \frac{2T}{1-16\kappa}\left(\frac{\mathcal{L}\eta}{m} + 4 + 9\mathcal{L}^2\right)\sigma_L^2 + \frac{540T\mathcal{L}^2\eta^2\tau^2}{1-16\kappa}\sigma_G^2$$

$$+ \frac{540\mathcal{L}^2\eta^2\tau^2}{1-16\kappa} \sum_{t=0}^{T-1} \mathbb{E}\left[\|\nabla F(\mathbf{x}_t)\|^2\right]$$

$$\leq \frac{2}{(1-16\kappa)\eta\tau}(F(\mathbf{x}_0) - F(\mathbf{x}_T)) + \frac{2T}{1-16\kappa}\left(\frac{\mathcal{L}\eta}{m} + 4 + 9\mathcal{L}^2\right)\sigma_L^2 + \frac{540T\mathcal{L}^2\eta^2\tau^2}{1-16\kappa}\sigma_G^2$$

$$+ \frac{1}{2} \sum_{t=0}^{T-1} \mathbb{E}\left[\|\nabla F(\mathbf{x}_t)\|^2\right] \tag{22}$$

$$\leq \frac{4}{(1-16\kappa)\eta\tau}(F(\mathbf{x}_0) - F(\mathbf{x}_T)) + \frac{4T}{1-16\kappa}\left(\frac{\mathcal{L}\eta}{m} + 4 + 9\mathcal{L}^2\right)\sigma_L^2 + \frac{1080T\mathcal{L}^2\eta^2\tau^2}{1-16\kappa}\sigma_G^2.$$

where (22) holds if $\eta \leq \frac{1-16\kappa}{6\sqrt{30}\mathcal{L}\tau}$. $\qquad\square$

## A.3 Experimental Settings in details

**Implementation Details** – All our experiments are conducted on a GPU cluster that contains 2 NVIDIA A6000 GPUs per machine. We use TensorFlow 2.15.0 for training and MPI for model aggregations. All individual experiments are performed at least three times, and the average accuracies are reported. The total number of clients is 128 and randomly chosen 32 clients participate in every communication round. We use mini-batch SGD with momentum (0.9) as the local optimizer. Table 6 shows the hyper-parameter settings for all our experiments, used not only for our method but also for other SOTA methods.

| Hyperparameters | CIFAR-10 (ResNet20) | CIFAR-100 (WRN-28) | FEMNIST (CNN) | AG News (DistillBERT) |
|---|---|---|---|---|
| $\tau$ (local steps) | 20 | 20 | 20 | 20 |
| batch size | 32 | 32 | 20 | 128 |
| min learning rate | 0.2 | 0.1 | 0.01 | 1e−5 |
| max learning rate | 0.2 | 0.4 | 0.01 | 1e−5 |
| total epoch | 200 | 300 | 200 | 100 |
| weight decay | 1e−4 | 1e−5 | 1e−4 | 1e−4 |
| decay epoch | 100, 150 | 150, 200 | 100, 150 | 60, 80 |

Table 6: Hyperparameter Settings for all experiments

**Artificial Data Heterogeneity** – For benchmark datasets that are not naturally non-IID, we generate artificial data distributions using Dirichlet's distributions. To evaluate the performance of our proposed method under realistic FL environments, the concentration coefficient $\alpha$ is configured as $0.1$ for CIFAR-10, CIFAR-100, and FEMNIST, and as $0.5$ for AG News. Note that these small concentration coefficient values represent highly heterogeneous distributions of local samples across clients as well as imbalance in the number of samples across labels.

**Algorithm-Specific Hyperparameter Selection** – Here, we summarize the hyper-parameter settings used to reproduce other SOTA methods, primarily following the configurations outlined in the original papers. We find algorithm-specific hyper-parameters using a grid search that achieve accuracy reasonably close to the baseline algorithm (FedAvg) while minimizing communication costs, and then measure the validation accuracy as shown in Section 4.1. Table 7 and Table 8 show the hyper-parameter settings for SOTA methods and experiments shown in Table 2 and 3, respectively. When running FedPara, both convolution layers and fully connected (FC) layers are re-parameterized using their proposed method. All hyperparameters shown in Table 7 and 8 are defined in the original papers.

| Algorithm | Hyperparameters | CIFAR-10 (ResNet20) | CIFAR-100 (WRN-28) | FEMNIST (CNN) | AG News (DistillBERT) |
|---|---|---|---|---|---|
| FedPAQ | $s$ (quantization level) | 16 | 16 | 8 | 8 |
| FedPara | parameters ratio [%] | 0.5 | 0.6 | 0.2 | 0.3 |
| LBGM | $\delta$ (threshold) | 0.95 | 0.98 | 0.96 | 0.6 |
| PruneFL | reconfiguration iteration | 50 | 50 | 50 | 50 |
| FedDropoutAvg | $fdr$ (federated dropout rate) | 0.5 | 0.4 | 0.75 | 0.5 |
| FedBAT | $\rho, \phi$ (coefficient, warm-up ratio) | 6, 0.5 | 6, 0.5 | 6, 0.5 | 6, 0.5 |

Table 7: Hyperparameter Settings for Comparative Study 4.1 of communication-efficient FL methods

| Algorithm | Hyperparameters | CIFAR-10 (ResNet20) | FEMNIST (CNN) |
|---|---|---|---|
| FedProx | $\mu$ (proximal term coefficient) | 0.001 | 0.001 |
| FedPAQ | $s$ (quantization level) | 16 | 8 |
| FedOpt | $\eta$ (server learning rate) | 0.9 | 1.2 |
| MOON | $\mu$ (control the weight of model-contrastive loss), $\tau$ (temperature parameter) | 1, 1.5 | 1, 0.5 |
| FedMut | $\alpha$ (distance scaling factor), $\beta$ (dynamic mutation factor) | 0.5, 1 | 0.5, 1 |
| FedACG | $\lambda$ (global momentum scaling factor), $\beta$ (penalty coefficient) | 0.7, 0.01 | 0.7, 0.01 |

Table 8: Hyperparameter Settings for Harmonization with Other FL methods 4.2

## A.4 Extra Results

**Sensitivity on** $\delta$ – We performed a grid search for each dataset to find the best $\delta$ setting which yields reasonable accuracy together with the maximum communication cost reduction. Table 9–12 show the four benchmarks' accuracy and communication costs corresponding to various $\delta$ settings.

**How much could be recycled safely?** – We also investigate the impact of $\delta$ on model accuracy and communication costs. We divide the number of model aggregations at each layer by the total number of communication rounds to calculate the layer-wise communication cost. Then, we sum up the calculated layer-wise costs to get the total communication cost. Intuitively, the larger the $\delta$, the lower the communication cost. However, the model accuracy is expected to drop as more layers have their updates recycled.

Table 9 and 10 show CIFAR-10 and CIFAR-100 experimental results for various $\delta$ settings. One key observation is that the accuracy is almost not degraded when LUAR is applied with $\delta \leq 12$ for both datasets. This means that many network layers have quite stable gradient dynamics, and thus their updates can be safely recycled. In addition, the accuracy is hardly reduced until the communication cost is reduced by almost $50\%$. This is a significant benefit especially in FL environments where the network bandwidth is extremely limited.

**Results under varying degrees of data heterogeneity** – The degree of non-IIDness strongly affects the training efficiency of Federated Learning methods. We conducted additional experiments to demonstrate that FedLUAR is robust to various degrees of non-IIDness. Table 13 and Table 14 show CIFAR-10 and AG News experimental results for various Dirchlet concentration factor $\alpha$ settings. In both benchmarks, FedLUAR achieves comparable accuracy to FedAvg regardless of $\alpha$, while considerably reducing the communication cost. Therefore, we conclude that FedLUAR is robust to the degree of non-IIDness.

**Performance under different numbers of active clients** – We conducted additional ablation study using different numbers of active clients. Table 15 and Table 16 show that, regardless of the total number of clients, FedLUAR achieves accuracy comparable to FedAvg while significantly reducing the communication cost. This ablation study demonstrates the superior scalability of FedLUAR.

**Learning Curves** – Figure 5 and 6 show the learning curve comparisons for CIFAR-100 and FEMNIST benchmarks, respectively. To highlight the difference clearly, we chose only 3 representative methods and compare their curves to those of FedLUAR. It is clearly shown that FedLUAR achieves virtually the same accuracy as FedAvg while having a significantly reduced communication cost. These results well prove the efficacy of the proposed update recycling method.

| $\delta$ | Validation Accuracy (%) | Communication Cost |
|---|---|---|
| 0 | $61.27 \pm 0.7\%$ | 1.00 |
| 4 | $61.25 \pm 0.4\%$ | 0.84 |
| 8 | $60.92 \pm 1.7\%$ | 0.68 |
| 12 | $60.15 \pm 0.7\%$ | 0.47 |
| 16 | $50.07 \pm 1.6\%$ | 0.30 |

Table 9: The CIFAR-10 (ResNet20) classification performance with varying $\delta$ settings.

| $\delta$ | Validation Accuracy (%) | Communication Cost |
|---|---|---|
| 0 | $59.88 \pm 0.8\%$ | 1.00 |
| 4 | $59.85 \pm 0.1\%$ | 0.88 |
| 8 | $59.93 \pm 0.1\%$ | 0.76 |
| 12 | $59.73 \pm 0.6\%$ | 0.61 |
| 14 | $56.49 \pm 0.1\%$ | 0.54 |
| 16 | $55.03 \pm 0.7\%$ | 0.51 |
| 20 | $49.60 \pm 0.2\%$ | 0.36 |

Table 10: The CIFAR-100 (WideResNet28) classification performance with varying $\delta$ settings.

| $\delta$ | Validation Accuracy (%) | Communication Cost |
|---|---|---|
| 0 | $71.01 \pm 0.4\%$ | 1.00 |
| 1 | $71.46 \pm 0.1\%$ | 0.50 |
| 2 | $73.17 \pm 1.1\%$ | 0.18 |
| 3 | $60.35 \pm 2.6\%$ | 0.03 |

Table 11: The FEMNIST (CNN) classification performance with varying $\delta$ settings.

| $\delta$ | Validation Accuracy (%) | Communication Cost |
|---|---|---|
| 0 | $82.66 \pm 0.1\%$ | 1.00 |
| 10 | $82.82 \pm 0.1\%$ | 0.56 |
| 20 | $82.24 \pm 0.1\%$ | 0.36 |
| 30 | $82.80 \pm 0.1\%$ | 0.17 |
| 35 | $79.00 \pm 0.1\%$ | 0.08 |

Table 12: The AG news (DistillBERT) classification performance with varying $\delta$ settings.

| Method | | $\alpha = 0.1$ | $\alpha = 0.5$ | $\alpha = 1.0$ |
|---|---|---|---|---|
| | Comm | Acc | Acc | Acc |
| FedAvg | 1.00 | 61.27% | 76.54% | 79.73% |
| FedLUAR | 0.47 | 60.15% | 76.04% | 79.50% |

Table 13: CIFAR-10 with various Dirchlet concentration factor $\alpha$ settings. The number of recycled layers, $\delta = 10$ out of 20 layers in ResNet20.

| Method | | $\alpha = 0.1$ | $\alpha = 0.5$ | $\alpha = 1.0$ |
|---|---|---|---|---|
| | Comm | Acc | Acc | Acc |
| FedAvg | 1.00 | 81.53% | 82.66% | 83.22% |
| FedLUAR | 0.17 | 81.88% | 82.80% | 82.75% |

Table 14: AG News with various Dirchlet concentration factor $\alpha$ settings. The number of recycled layers, $\delta = 30$ out of 40 layers in DistilBERT.

| Method | | 64 (0.5) | 128 (0.25) | 256 (0.125) |
|---|---|---|---|---|
| | Comm | Acc | Acc | Acc |
| FedAvg | 1.00 | 54.24% | 61.27% | 57.81% |
| FedLUAR | 0.48 | 53.34% | 60.15% | 57.81% |

Table 15: The $\delta = 10$ out of 20 layers in ResNet20 for FedLUAR. 64, 128, and 256 indicate the total number of clients, and the numbers in parentheses (0.5, 0.25, 0.125) represent the client activation ratio on CIFAR-10.

| Method | | 64 (0.5) | 128 (0.25) | 256 (0.125) |
|---|---|---|---|---|
| | Comm | Acc | Acc | Acc |
| FedAvg | 1.00 | 66.31% | 71.01% | 75.97% |
| FedLUAR | 0.14 | 68.15% | 73.17% | 76.72% |

Table 16: The $\delta = 2$ out of 4 layers in CNN for FedLUAR. 64, 128, and 256 indicate the total number of clients, and the numbers in parentheses (0.5, 0.25, 0.125) represent the client activation ratio on FEMNIST.

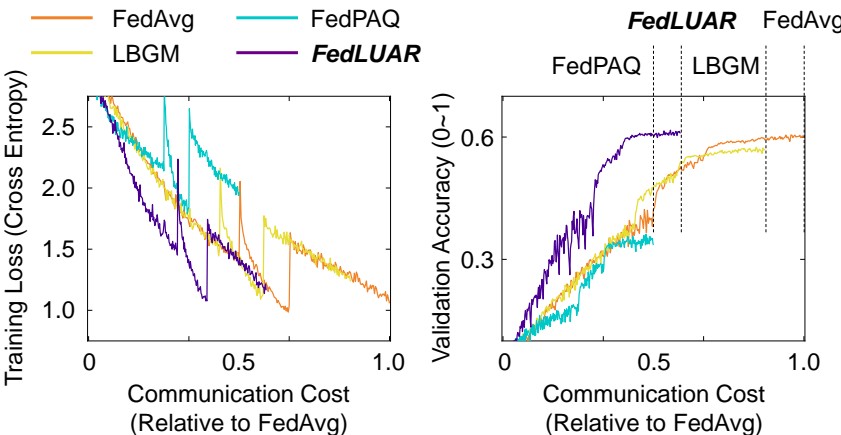

Figure 5: The learning curve comparisons for CIFAR-100 (Wide-ResNet28-10). The x-axis represents the communication cost relative to FedAvg. FedPAQ has the least amount of communication cost for 300 epochs, however it loses the accuracy too much. `FedLUAR` nearly does not drop the accuracy while significantly reducing the communication cost.

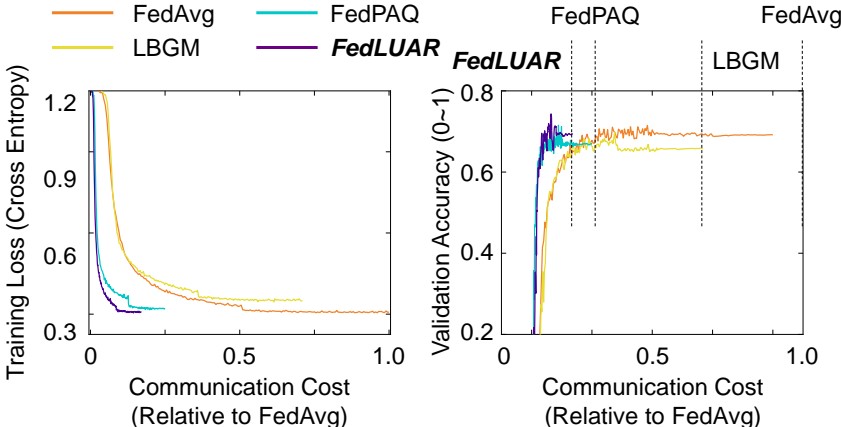

Figure 6: The learning curve comparisons for FEMNIST (CNN). The x-axis represents the communication cost relative to FedAvg. `FedLUAR` significantly reduces the communication cost while maintaining the model accuracy.

