# OpenReview forum: "Layer-wise Update Aggregation with Recycling for Communication-Efficient Federated Learning"
_NeurIPS.cc/2025/Conference — NeurIPS 2025 poster_

### Official Review · Reviewer_eMSU · 2025-06-01

**Clarity:** 3
**Significance:** 3
**Originality:** 3
**Rating:** 4
**Confidence:** 5

**Summary:**

This paper proposes FedLUAR, a novel federated learning method that recycles previous model updates on selected layers based on a gradient-to-weight ratio metric. The approach is motivated by the need to reduce communication costs in federated learning without sacrificing accuracy, and it challenges conventional methods that drop updates. Extensive theoretical analysis and empirical evaluations on benchmark datasets demonstrate that the layer-wise update recycling mechanism can significantly reduce communication overhead while maintaining or even accelerating model convergence.

**Questions:**

See weaknesses.

**Ethical Concerns:**

["NO or VERY MINOR ethics concerns only"]

**Limitations:**

Limitations are mentioned in the text as the extra memory overhead.

**Quality:**

3

**Strengths And Weaknesses:**

Strengths:

1. The paper introduces an innovative update recycling strategy that selectively reuses past layer updates instead of dropping them, offering a fresh perspective on reducing communication costs in federated learning. The method leverages a newly defined metric based on the gradient-to-weight ratio to prioritize layers effectively.

2. The proposed FedLUAR framework achieves significant communication efficiency while preserving model accuracy as demonstrated by experiments on multiple datasets, including image and text classification tasks. The empirical results show that the method reduces communication cost. The experiments are comprehensive and include comparisons with several state-of-the-art communication-efficient federated learning methods.

3. The paper provides rigorous theoretical analysis including convergence proofs and noise bounding under realistic assumptions. The theoretical section clearly lays out the assumptions, lemmas, and theorems necessary for understanding how the update recycling mechanism maintains the stability of model updates. The proofs are detailed enough to give confidence in the method’s practical applicability in non-convex settings.

4. Extensive ablation studies are performed to validate the effectiveness of the proposed layer prioritization metric and the recycling scheme. The ablation experiments compare different layer selection strategies and reinforce the idea that the gradient-to-weight ratio is a strong indicator for identifying less critical layers. The experiments also systematically explore the impact of key hyperparameters on both communication cost and model performance.


Weaknesses:

1. The method introduces additional memory overhead on the server as previous updates for certain layers must be stored, which may become a bottleneck for models with very deep or large architectures. The paper does mention the extra memory requirement but does not quantify its impact extensively. The discussion leaves open questions on scalability when server resources are very limited.

2. The approach relies on the tuning of a key hyperparameter (δ) that determines the number of layers to recycle, and optimal settings appear to be dataset dependent. The ablation results indicate that performance can drop if recycling is applied excessively, suggesting sensitivity to this parameter. There is limited guidance on how to best select δ in various federated learning scenarios in real-world deployments.

3. Although the experiments are extensive, the evaluation is limited to a handful of benchmark datasets and standard models, leaving some uncertainty about how the approach will perform in more diverse or larger-scale federated learning environments. The paper does not include scenarios with extremely heterogeneous data distributions that might reveal additional challenges. Information on performance in more varied real-world applications is thus lacking.

4. Some highly related works about compression in FL are not discussed. [1,2] proposed compressing the communicated parameters or the model delta. [2,3] proposed using SVD to compress model parameters. [5] can be regarded as a layer-wise model compression methods.


[1] Z. Tang, S. Shi, B. Li and X. Chu, "GossipFL: A Decentralized Federated Learning Framework With Sparsified and Adaptive Communication," In IEEE Transactions on Parallel and Distributed Systems. 2022

[2] Z. Tang, J. Huang, R. Yan, Y. Wang, Z. Tang, S. Shi, A.C. Zhou, X. Chu. Bandwidth-Aware and Overlap-Weighted Compression for Communication-Efficient Federated Learning. In Proceedings of the 53rd International Conference on Parallel Processing, 2024.

[3] Chai, Di, et al. "Practical lossless federated singular vector decomposition over billion-scale data." Proceedings of the 28th ACM SIGKDD, 2022.

[4] Haolin Wang, Xuefeng Liu, Jianwei Niu, Shaojie Tang. SVDFed: Enabling Communication-Efficient Federated Learning via Singular-Value-Decomposition. In IEEE INFOCOM 2023.

[5] H Wang, M Yurochkin, Y Sun, D Papailiopoulos, Y Khazaeni. Federated learning with matched averaging. In ICLR 2020.

---

> ### Author Rebuttal · Authors · 2025-07-28
>
> We thank the reviewer for their time and effort. Below are our responses and clarifications for each of the reviewer's questions.
>
> **Q1**: The paper does mention the extra memory requirement but does not quantify its impact extensively.
>
> **A1**: We would like to clarify that ***FedLUAR*’s memory footprint is actually smaller than that of FedAvg** (We realized the misanalysis after submitting the paper). Since the global model should be held in the memory space regardless of which algorithm is used, we only focus on the local models.
> - In FedAvg, server should receive local models from all active clients. Thus, the memory footprint is $a \cdot d$, where a is the number of active clients and d is the model size.
> - *FedLUAR* receives local models from active clients except \delta layers. Thus, the memory footprint is $a \cdot (d - k)$, where $k$ is the size of $\delta$ layers. Instead, the previous global update should be kept in the memory space for the $\delta$ layers, consuming $k$ space only. Therefore, *FedLUAR*’s memory footprint is $a \cdot (d - k) + k < a \cdot d$.
>
> To support our analysis, we actually measured the memory footprint of FedAvg and *FedLUAR* during training. First, the total number of clients is 128 and only randomly selected 32 clients are activated at each communication round. We use MPI to run FL on 2 GPUs. Thus, each process locally train 16 models and then all the locally trained models are aggregated using *MPI_Allreduce()*. Under this setting, we measured the memory footprint of each process, observed during training.
>
> | CIFAR-10 (ResNet20) | $\delta$ | Memory Footprint (MB) |
> | :--- | :---:|:---:|
> |FedAvg| - | 33.49 |
> | *FedLUAR* | 10 | 15.23 |
>
> | CIFAR-100 (WRN28-10) | $\delta$ |  Memory Footprint (MB) |
> | :--- | :---:|:---:|
> |FedAvg | - | 4462.80 |
> |*FedLUAR* | 14 | 2604.88|
>
> | FEMNIST (CNN) | $\delta$ |  Memory Footprint (MB) |
> | :--- | :---:|:---:|
> |FedAvg | - | 806.11 |
> |*FedLUAR* | 2 | 204.73 |
>
> | AG News (DistillBERT) | $\delta$ |  Memory Footprint (MB) |
> | :--- | :---:|:---:|
> |FedAvg | - | 8294.18 |
> |*FedLUAR* | 30 | 1825.42|
>
> We clearly see that *FedLUAR* actually consumes less memory space than FedAvg. Thank you again for pointing out this. We will add a brief discussion on this memory consumption in Section 3.2 and Appendix.
>
> --
>
> **Q2**: There is limited guidance on how to best select $\delta$ in various federated learning scenarios in real-world deployments.
>
> **A2**: Thank you for this sharp feedback. We acknowledge that our paper provides only limited guidance on how to find good $\delta$ values. As pointed out by reviewer, the best setting of $\delta$ varies depending on many factors including training data distribution, model size, and non-IIDness. In our extensive empirical study, however, we find that the method generally performs well when $\delta$ is not larger than $\sim 63$% of the total number of layers (averaged across all the benchmarks). A more exhaustive ablation study would help readers easily identify good hyperparameter settings. We plan to extend this work to address system heterogeneity and will include such an ablation study in future work.
>
> --
>
> **Q3**: The paper does not include scenarios with extremely heterogeneous data distributions that might reveal additional challenges.
>
> **A3**: We used Dirichlet concentration parameter of $0.1$ for artificial non-IID datasets. We also used unbalanced distributions with respect to classes. For instance, each local dataset of CIFAR-10 contains up to 3 classes only. This setting represents highly non-IID environments. Considering that many previous works use larger concentration parameter values such as $\alpha=0.5$, $0.8$, or $1.0$, we believe our experimental settings are sufficiently challenging/extreme.
>
> Nevertheless, we conducted additional experiments to further provide empirical evidence demonstrating the effectiveness of *FedLUAR*, as summarized below.
>
> - CIFAR-10 (ResNet20) with various Dirchlet concentration factor $\alpha$ settings. The number of recycled layers, $\delta = 10$ out of $20$ layers in ResNet20.
>     |   Method   |  | $\alpha=0.1$ | $\alpha=0.5$ | $\alpha=1.0$ |
>     |:----------:|:-----:|:------------:|:------------:|:------------:|
>     |            |  Comm   |     Acc      |     Acc      |     Acc      |
>     |  FedAvg    | 1.00  |   61.27%     |   76.54%     |   79.73%     |
>     | *FedLUAR*  | 0.47  |   60.15%     |   76.04%     |   79.50%     |
>
> - AG News (DistillBERT) with various Dirichlet concentration factor $\alpha$ settings. The number of recycled layers, $\delta = 30$ out of $40$ layers in DistillBERT.
>     |   Method   |  | $\alpha=0.2$ | $\alpha=0.5$ | $\alpha=0.8$ |
>     |:----------:|:-----:|:------------:|:------------:|:------------:|
>     |            |  Comm   |     Acc      |     Acc      |     Acc      |
>     |  FedAvg    | 1.00  |   81.53%     |  82.66%      |   83.22%     |
>     | *FedLUAR*  | 0.17  |   81.88%     |   82.80%     |   82.75%     |
>
> In both benchmarks, *FedLUAR* achieves comparable accuracy to FedAvg regardless of $\alpha$, while considerably reducing the communication cost. Therefore, we conclude that ***FedLUAR* is robust to the degree of non-IIDness**.
>
> --
>
> **Q4**: Some highly related works about compression in FL are not discussed.
>
> **A4**: Thank you for bringing these related works to our attention. We will include a discussion of these prior works in Section 2, highlighting their respective strengths and limitations. We expect our update recycling method to be compatible with those compression-based FL methods, as it has already been empirically shown to work with both structured and sketch-based compression approaches such as FedPAQ and PruneFL.
>
> To further support our argument, we performed extra experiments to compare *FedLUAR* and PruneFL, a sparsification-based FL method.
>  - CIFAR-10 (ResNet20) with PruneFL. The number of recycled layers, $\delta = 10$ out of $20$ layers in ResNet20.
> | Method | Accuracy | Comm |
> | :--- | :---: | :---: |
> | PruneFL | 56.76% | 0.69|
> | PruneFL + *LUAR* | 55.43% | 0.49 |
>
>  - FEMNIST (CNN) with PruneFL. The number of recycled layers, $\delta = 2$ out of $4$ layers in a CNN.
> | Method | Accuracy | Comm |
> | :--- | :---: | :---: |
> | PruneFL | 69.42% | 0.19|
> | PruneFL + *LUAR* | 69.11% | 0.11 |
>
> In both benchmarks, we see that our proposed update recycling method maintains the accuracy while further reducing the communication cost of PruneFL. So, we conclude that *FedLUAR* is complementary to sparsification methods.

---

> > ### Comment · Reviewer_eMSU · 2025-08-01
> >
> > Thanks for the rebuttal. My previous concerns are addressed. I will keep my score.

---

### Official Review · Reviewer_ocSH · 2025-06-26

**Clarity:** 3
**Significance:** 3
**Originality:** 3
**Rating:** 4
**Confidence:** 2

**Summary:**

This paper proposes FedLUAR, a Layer-wise Update Aggregation with Recycling scheme for communication-efficient federated learning. FedLUAR selects a subset of layers based on a proposed metric that quantifies how aggregated gradients influence model parameter values in each layer, and recycles previous updates on the server side rather than discarding them.

**Questions:**

As stated in weakness.
Further experiments suggestions:
- Experiments examining how varying degrees of data heterogeneity affect the proposed method's performance would strengthen the evaluation.
- Evaluation with different numbers of participating users to understand scalability limitations and performance trade-offs.
- Investigation of the method's benefits in few-shot learning scenarios, if applicable to the problem domain.
- Experiments on larger datasets and modern foundation models to demonstrate practical applicability.

**Ethical Concerns:**

["NO or VERY MINOR ethics concerns only"]

**Final Justification:**

This rebuttal addresses most of my concerns. I'll keep my score.

**Limitations:**

Yes.

**Paper Formatting Concerns:**

In appendix, Several equations in the appendix exceed page width and require reformatting.

Also, appendix contains excessive whitespace that should be addressed for better presentation.

**Quality:**

3

**Strengths And Weaknesses:**

Strength:
- The concept of recycling previous updates rather than simply dropping them presents an interesting approach to improving communication efficiency in federated learning.

Weakness:
- The paper lacks clear theoretical justification for the Gradient-Weight Ratio Analysis. Is there an optimal proof or empirical motivation for this metric, or does it serve primarily as a heuristic function? The theoretical grounding needs strengthening.
- As acknowledged in the limitations section, FedLUAR requires storing weights from all users in previous communication rounds, leading to significant memory overhead. This limitation may prevent the method from scaling to scenarios with large numbers of users training large models.
- The main contributions of this work should be more explicitly stated and clearly distinguished from existing methods.
- Beyond YOGA, the paper should include comparisons with other layer-wise model aggregation methods for federated learning with centralized servers. A comprehensive comparison would better position this work within the existing literature.

---

> ### Author Rebuttal · Authors · 2025-07-29
>
> We thank the reviewer for the valuable comments. Please find our responses to each of the reviewer's questions.
>
> **Q1**: Experiments examining how varying degrees of data heterogeneity affect the proposed method's performance would strengthen the evaluation.
>
> **A1**: As pointed out by the reviewer, yes, the degree of non-IIDness strongly affects the training efficiency of FL methods. We conducted additional experiments to demonstrate that *FedLUAR* is robust to the degree of non-IIDness as follows.
>
> - CIFAR-10 (ResNet20) with various Dirchlet concentration factor $\alpha$ settings. The number of recycled layers, $\delta = 10$ out of $20$ layers in ResNet20.
>     |   Method   |  | $\alpha=0.1$ | $\alpha=0.5$ | $\alpha=1.0$ |
>     |:----------:|:-----:|:------------:|:------------:|:------------:|
>     |            |  Comm   |     Acc      |     Acc      |     Acc      |
>     |  FedAvg    | 1.00  |   61.27%     |   76.54%     |   79.73%     |
>     | *FedLUAR*  | 0.47  |   60.15%     |   76.04%     |   79.50%     |
>
> - AG News (DistillBERT) with various Dirichlet concentration factor $\alpha$ settings. The number of recycled layers, $\delta = 30$ out of $40$ layers in DistillBERT.
>     |   Method   |  | $\alpha=0.2$ | $\alpha=0.5$ | $\alpha=0.8$ |
>     |:----------:|:-----:|:------------:|:------------:|:------------:|
>     |            |  Comm   |     Acc      |     Acc      |     Acc      |
>     |  FedAvg    | 1.00  |   81.53%     |  82.66%      |   83.22%     |
>     | *FedLUAR*  | 0.17  |   81.88%     |   82.80%     |   82.75%     |
>
>
> In both benchmarks, *FedLUAR* achieves comparable accuracy to FedAvg regardless of $\alpha$, while considerably reducing the communication cost. Therefore, we conclude that ***FedLUAR* is robust to the degree of non-IIDness**.
>
> --
>
> **Q2**: Evaluation with different numbers of participating users to understand scalability limitations and performance trade-offs.
>
> **A2**: Thank you for this sharp comment. We conducted additional ablation study with various numbers of active clients as follows.
>
>  - CIFAR-10 (ResNet20) experiments. The $\delta=10$ for *FedLUAR*. In the following table, 64, 128, and 256 indicate the total number of clients, and the numbers in parentheses (0.5, 0.25, 0.125) represent the client activation ratio.
> | | | 64 (0.5) | 128 (0.25) | 256 (0.125) |
> |---------: |----: |--------: |----------: |-----------: |
> |  Method   | Comm |  Acc |    Acc |     Acc |
> | FedAvg    | 1.00 |   54.24% |     61.27% |      58.07% |
> | *FedLUAR* | 0.48 |   53.34% |     60.15% |      57.81% |
>
>  - FEMNIST (CNN) experiments. The $\delta=2$ for *FedLUAR*. In the following table, 64, 128, and 256 indicate the total number of clients, and the numbers in parentheses (0.5, 0.25, 0.125) represent the client activation ratio.
> | | | 64 (0.5) | 128 (0.25) | 256 (0.125) |
> |---------: |----: |--------: |----------: |-----------: |
> |  Method   | Comm |  Acc |    Acc |     Acc |
> | FedAvg    | 1.00 |   66.31% |     71.01% |      75.97% |
> | *FedLUAR* | 0.14 |   68.15% |     73.17% |      76.72% |
>
> We clearly see that, regardless of the total number of clients, *FedLUAR* matches the accuracy of FedAvg while remarkably reducing the communication cost. This ablation study demonstrates the superior scalability of *FedLUAR*. We will add this discussion to Appendix.
>
> --
>
> **Q3**: Investigation of the method's benefits in few-shot learning scenarios, if applicable to the problem domain.
>
> **A3**:  Thank you for suggesting an interesting application of our method: update recycling for efficient federated few-shot learning. We consider this a promising direction for future work. Although we are unable to implement and evaluate it within the limited rebuttal period, we will certainly explore it in future research.
>
> --
>
> **Q4**: Experiments on larger datasets and modern foundation models to demonstrate practical applicability.
>
> **A4**: We recognize that the machine learning community is currently highly interested in efficient training of large foundation models. Our experimental results in Tables 1, 3, and 4 include the AG News benchmark conducted using DistilBERT, a representative modern foundation model. We believe that these results and our analysis offer useful insights into communication-efficient large-scale federated learning for the broader ML community.
>
> --
>
> Finally, we would like to clarify that ***FedLUAR*’s memory footprint is actually smaller than that of FedAvg** (We realized the misanalysis after submitting the paper). Since the global model should be held in the memory space regardless of which algorithm is used, we only focus on the local models.
> - In FedAvg, server should receive local models from all active clients. Thus, the memory footprint is $a \cdot d$, where a is the number of active clients and d is the model size.
> - *FedLUAR* receives local models from active clients except \delta layers. Thus, the memory footprint is $a \cdot (d - k)$, where $k$ is the size of $\delta$ layers. Instead, the previous global update should be kept in the memory space for the $\delta$ layers, consuming $k$ space only. Therefore, *FedLUAR*’s memory footprint is $a \cdot (d - k) + k < a \cdot d$.
>
> To support our analysis, we actually measured the memory footprint of FedAvg and *FedLUAR* during training. First, the total number of clients is 128 and only randomly selected 32 clients are activated at each communication round. We use MPI to run FL on 2 GPUs. Thus, each process locally train 16 models and then all the locally trained models are aggregated using *MPI_Allreduce()*. Under this setting, we measured the memory footprint of each process, observed during training.
>
> | CIFAR-10 (ResNet20) | $\delta$ | Memory Footprint (MB) |
> | :--- | :---:|:---:|
> |FedAvg| - | 33.49 |
> | *FedLUAR* | 10 | 15.23 |
>
> | CIFAR-100 (WRN28-10) | $\delta$ |  Memory Footprint (MB) |
> | :--- | :---:|:---:|
> |FedAvg | - | 4462.80 |
> |*FedLUAR* | 14 | 2604.88|
>
> | FEMNIST (CNN) | $\delta$ |  Memory Footprint (MB) |
> | :--- | :---:|:---:|
> |FedAvg | - | 806.11 |
> |*FedLUAR* | 2 | 204.73 |
>
> | AG News (DistillBERT) | $\delta$ |  Memory Footprint (MB) |
> | :--- | :---:|:---:|
> |FedAvg | - | 8294.18 |
> |*FedLUAR* | 30 | 1825.42|
>
> We clearly see that *FedLUAR* actually consumes less memory space than FedAvg. Thank you again for pointing out this. We will add a brief discussion on this memory consumption in Section 3.2 and Appendix.
>
> Finally, we revised Introduction section to clearly summarize our main contributions from this research work.

---

> > ### Comment · Reviewer_ocSH · 2025-08-01
> >
> > Thank you for your rebuttal. This rebuttal addresses most of my concerns. I'll keep my score.

---

> > > ### Author Response · Authors · 2025-08-06
> > > **Mandatory acknowledgment**
> > >
> > > Thank for checking out our rebuttal and your response.
> > > Could you please push the "Mandatory Acknowledgment" button to finalize the discussion?
> > > I believe this should be done by all individual reviewers.

---

### Official Review · Reviewer_sW79 · 2025-06-30

**Clarity:** 2
**Significance:** 2
**Originality:** 2
**Rating:** 5
**Confidence:** 3

**Summary:**

The paper introduces a communication-efficient approach to distributed training by selectively reusing updates for certain layers, avoiding aggregation in every round. Layer selection is guided by the gradient-to-weight ratio, in contrast to prior work that considers only gradient magnitude or weight norms. Experiments reveal that layers with large gradients may also have large weights, making their updates less impactful. Theoretical analysis bounds the noise introduced by partial aggregation, while empirical results demonstrate improvements in model performance, communication efficiency, and layer criticality metrics.

**Questions:**

1 -    On which side is the computation of layer importance performed? When only a subset of parameters is shared, does the selection happen on the client side before sending, or is it determined by the server?

2 -    What does \kappa refer to in Remark 1 (line 222)? It appears without a clear definition.

3 -    Remark 2 (line 226) states that all terms on the right-hand side of inequality (7) vanish with a diminishing learning rate, except part of the second term. However, the first term seems to contain the learning rate in its discriminant - could you clarify?

4 -    The evaluation and the last paragraph of Section 3.3 discuss data heterogeneity, although the method itself is not specifically designed for it. Why is heterogeneity relevant in this context?

**Ethical Concerns:**

["NO or VERY MINOR ethics concerns only"]

**Final Justification:**

The rebuttal clarified my concerns and did not change my score decision.

**Limitations:**

N\A

**Paper Formatting Concerns:**

N\A

**Quality:**

3

**Strengths And Weaknesses:**

The observation on balancing weight norms with gradient norms is insightful and, according to empirical results, contributes to improved distributed training.

However, the algorithm description lacks clarity regarding where computations occur: criticality scores are computed on the server, yet clients require them to determine which parameters to share (line 181).

While the theoretical analysis provides bounds on the noise under a diminishing learning rate, it does not offer justification for the critical layer selection strategy.

Empirical evaluation is not taking as benchmark method in [1].

[1] Kamp M, Adilova L, Sicking J, Hüger F, Schlicht P, Wirtz T, Wrobel S. Efficient decentralized deep learning by dynamic model averaging. InMachine Learning and Knowledge Discovery in Databases: European Conference, ECML PKDD 2018, Dublin, Ireland, September 10–14, 2018, Proceedings, Part I 18 2019 (pp. 393-409). Springer International Publishing.

---

> ### Author Rebuttal · Authors · 2025-07-28
>
> We appreciate the reviewer’s careful and thorough comments. Please find our responses to each of the reviewer's questions.
>
>
> **Q1**: On which side is the computation of layer importance performed?
>
> **A1**: As shown in Algorithm 1 (*LUAR*), once the local models are aggregated at the server, the **server computes the layer-wise importance** (line 6). It then sends minimal information to the clients, indicating which layers will be recycled. Based on that information, clients skip uploading those layers to the server at the end of the communication round. Therefore, *FedLUAR* does not impose any additional computational overhead on the clients for calculating layer importance.
>
> --
>
> **Q2**: What does $\kappa$ refer to in Remark 1 (line 222)?
>
> **A2**: Thank you for pointing out this presentation quality issue. Actuatlly, $\kappa$ is defined at line 208. This is the discrepancy between the magnitude of FedAvg’s update and that of FedLUAR’s update. In Remark 1, “small $\kappa$” indicates that the discrepancy is small (updates are recycled in fewer layers).
>
> --
>
> **Q3**: How does the first term on the right-hand side of inequality (7) vanish?
>
> **A3**: We acknowledge that Remark 2 is written in an unclear way. *FedLUAR* guarantees convergence as $T \rightarrow \infty$. For simplicity, we omitted $1/T$ on both sides in (7). If we multiply $1/T$ on both sides and $T \rightarrow \infty$, the first term on the right-hand side vanishes (finite horizon result). We will accordingly revise the manuscript to clarify this.
>
> --
>
> **Q4**: Why is heterogeneity relevant in this context (the last paragraph of Section 3.3)?
>
> **A4**: We appreciate this critical question. We found out that our explanation was not clear enough. Please find the revised paragraph below. We hope this clarification addresses the reviewer’s concern.
>
> *In general, as the degree of non-IIDness increases, the global variance $\sigma_G$ will likely grow due to greater discrepancies among local datasets. As shown in the final term on the right-hand side of (7), recycling updates in more layers increases the coefficient, which in turn amplifies the final term. Consequently, the model is expected to converge more slowly.*
>
> **Extra response**: Additionally, we added a brief discussion on the comparison between *FedLUAR* and the recommended related work [1]. Thank you again for bringing this work to our attention. Although we could not implement and evaluate the dynamic model aggregation method due to the limited rebuttal period, we plan to include a comparative study incorporating it in our future extension of this work.

---

> > ### Comment · Reviewer_sW79 · 2025-08-01
> >
> > I thank the authors for the reply. It clarified my concerns.

---

### Official Review · Reviewer_21T7 · 2025-07-01

**Clarity:** 3
**Significance:** 3
**Originality:** 4
**Rating:** 5
**Confidence:** 4

**Summary:**

This paper introduces FedLUAR, a novel communication-efficient Federated Learning (FL) method designed to mitigate the high communication costs associated with training large models. The central idea is to reduce the volume of data transmitted from clients to the server by selectively "recycling" historical updates for certain model layers. The method proposes a layer prioritization metric, $s_{t,l}$, which is the ratio of a layer's aggregated gradient magnitude to its weight magnitude. This metric identifies layers where an update is likely to have a less substantial impact. In each communication round, FedLUAR uses a weighted random sampling scheme to select a subset of these low-priority layers. For the selected layers, clients do not transmit their newly computed updates; instead, the server re-applies the aggregated update from the previous communication round. For all other layers, training proceeds as in standard FedAvg. The authors provide a theoretical convergence analysis and conduct extensive experiments on CIFAR-10/100, FEMNIST, and AG News, demonstrating that FedLUAR can achieve comparable accuracy to FedAvg while reducing communication costs by over 80%. Furthermore, they show that the recycling mechanism is compatible with other advanced FL optimizers like FedProx and MOON.

**Questions:**

1.  Adaptive $\delta$: The number of recycled layers, $\delta$, is a crucial hyperparameter. Have you considered an adaptive schedule for $\delta$ that might, for example, decrease the number of recycled layers as training progresses and the model fine-tunes? A dynamic approach could potentially yield a better and more robust trade-off between communication and accuracy.

2.  Impact of Data Heterogeneity: The paper notes that convergence is guaranteed under non-IID conditions. Could you provide a deeper analysis or intuition on the interplay between the degree of data heterogeneity and the effectiveness of FedLUAR? Specifically, as non-IIDness increases, client updates diverge more. Does this make the *previous global update* a poorer proxy for the *current required update*, potentially slowing convergence more than in the IID case?

3.  Scalability to Large-Scale Models: Regarding the server-side memory required to store historical updates, could you provide a more formal analysis of its scaling behaviour? For a model with $L$ layers and parameters $\Theta$, the overhead appears to be $O(\sum_{l \in \text{recycled}} |\theta_l|)$. How would this fare with modern foundational models?

4.  Orthogonality with Update Compression: FedLUAR is shown to be compatible with different FL optimizers. Is the mechanism also orthogonal to other communication efficiency techniques like quantization or sparsification? For instance, could one apply 8-bit quantization to the non-recycled layer updates while recycling the others? An analysis of whether these benefits are additive would significantly strengthen the paper's positioning as a general-purpose tool.

**Ethical Concerns:**

["NO or VERY MINOR ethics concerns only"]

**Final Justification:**

The authors have adequately addressed my concerns, and I will maintain my original score.

**Limitations:**

The authors acknowledge some technical limitations related to the method's overhead, their treatment of broader limitations, particularly concerning privacy and fairness, is absent. Their discussion of negative societal impacts is also notably shallow, simply stating that none are anticipated and focusing exclusively on positive environmental benefits, which falls short of the comprehensive self-assessment expected for responsible AI research.

**Quality:**

3

**Strengths And Weaknesses:**

Strengths:

- Originality and Novelty: The core concept of recycling historical updates is a significant and novel contribution. While the field has extensively explored compression techniques like quantization, sparsification, and structured updates, the idea of reusing a previous update as a proxy for the current one is a fresh perspective. The ablation study, which demonstrates that recycling is substantially better than simply skipping the updates for low-priority layers, provides strong evidence for the value of this approach. This clearly distinguishes the work from prior art.

- Quality of the Proposed Metric: The paper proposes a nuanced layer-prioritization metric ($s_{t,l}$) that goes beyond simple gradient norms. By considering the update's magnitude relative to the layer's weights, it offers a more stable and potentially more accurate way to gauge an update's impact. This is a valuable insight, as it challenges the common assumption that larger gradients always correspond to more important updates. The empirical results appear to validate the effectiveness of this metric.

- Technical Soundness and Empirical Rigor: The claims are well-supported by both theoretical analysis and comprehensive experiments. The convergence analysis, which shows that the noise introduced by recycling is bounded under standard assumptions, provides a solid theoretical foundation. Empirically, the method is tested across multiple datasets, models, and degrees of data heterogeneity, consistently demonstrating significant communication savings with minimal to no loss in final accuracy.

- Significance and Practicality: The method's "plug-and-play" nature is a major strength. By demonstrating compatibility with established FL optimizers like FedProx and MOON, the authors position FedLUAR not as a competing algorithm but as a general-purpose enhancement. This significantly increases its potential for adoption by researchers and practitioners looking to make existing FL pipelines more efficient. The reported communication savings are substantial and practically meaningful.

Weaknesses:

- Unaddressed Server-Side Overheads: The paper claims the method requires "slightly more memory" on the server to store historical updates. This claim lacks quantification. For very large models (e.g., LLMs with billions of parameters), storing even a single round of global updates could be a significant memory burden, potentially becoming a new bottleneck. This aspect of practical feasibility needs a more thorough analysis.

- Hyperparameter Sensitivity: The performance of FedLUAR hinges on the hyperparameter $\delta$ (the number of recycled layers), which must be manually tuned. The paper does not provide clear guidance or an adaptive strategy for setting $\delta$, which could make it difficult to apply in practice. An effective trade-off between communication and accuracy depends on finding a near-optimal $\delta$, which may require a costly hyperparameter search for each new task.

Limited Evaluation in Dynamic Scenarios: The theoretical analysis assumes a standard FL setting. However, the core assumption that a stale update is a good proxy might break down in highly non-stationary environments where data distributions shift rapidly across rounds. In such cases, recycling could introduce significant error and slow down convergence. The practical impact in these more challenging (and realistic) scenarios is not sufficiently explored.

- Insufficient Comparison with Contemporary Layer-wise Methods: While the comparison to FedAvg is essential, the paper's positioning would be much clearer with a direct comparison to other recent layer-wise FL techniques. For instance, methods like FedLAMA [3, 4] also use layer-wise strategies (adaptive aggregation frequency) to improve efficiency. Understanding how FedLUAR's recycling mechanism compares to or complements such approaches is critical for assessing its state-of-the-art standing.

---

> ### Author Rebuttal · Authors · 2025-07-28
>
> We appreciate the reviewers’ thorough evaluations and positive feedback on our work. Below, we provide detailed responses to the reviewers’ comments and questions.
>
> **Q1**: Have you considered an adaptive schedule for \delta?
>
> **A1**: Thank you for the valuable suggestion. Yes, we agree that automating the choice of $\delta$ would make *FedLUAR* significantly more practical. We do consider deriving a robust way of finding the ‘safe’ number of recycled layers as a critical future work. In this work, we upper-bounded the noise introduced by update recycling in non-IID settings while preserving convergence properties. As a next step, we aim to develop a model-agnostic method for determining the number of recycled layers without violating the derived bound. Our research group is currently studying this topic to develop an efficient and automated layer-wise update recycling method.
>
> --
>
> **Q2**: Could you provide a deeper analysis or intuition on the interplay between the degree of data heterogeneity and the effectiveness of FedLUAR?
>
> **A2**: As pointed out by the reviewer, as the degree of non-IIDness increases, the convergence of *FedLUAR* becomes slower like any other FL methods. Specifically, in our convergence rate shown in (7), the final term with $\sigma_G$ on the right-hand side likely increases as the non-IIDness becomes stronger. Since the term has a complexity of $O(\frac{1}{1 - 16 \kappa})$, as $\kappa \in [0, \frac{1}{16})$ increases, the final term increases, making it converge more slowly. However, our analysis considers the maximum bound of $n_t$, the noise introduced by update recycling. Hence, the theoretical result is not affected by the degree of non-IIDness.
>
> We conducted additional experiments to demonstrate that *FedLUAR* is robust to the degree of non-IIDness in practice. Thank you for this sharp question. We will add discussion to Appendix.
>
> - CIFAR-10 (ResNet20) with various Dirchlet concentration factor $\alpha$ settings. The number of recycled layers, $\delta = 10$ out of $20$ layers in ResNet20.
>     |   Method   |  | $\alpha=0.1$ | $\alpha=0.5$ | $\alpha=1.0$ |
>     |:----------:|:-----:|:------------:|:------------:|:------------:|
>     |            |  Comm   |     Acc      |     Acc      |     Acc      |
>     |  FedAvg    | 1.00  |   61.27%     |   76.54%     |   79.73%     |
>     | *FedLUAR*  | 0.47  |   60.15%     |   76.04%     |   79.50%     |
>
> - AG News (DistillBERT) with various Dirichlet concentration factor $\alpha$ settings. The number of recycled layers, $\delta = 30$ out of $40$ layers in DistillBERT.
>     |   Method   |  | $\alpha=0.2$ | $\alpha=0.5$ | $\alpha=0.8$ |
>     |:----------:|:-----:|:------------:|:------------:|:------------:|
>     |            |  Comm   |     Acc      |     Acc      |     Acc      |
>     |  FedAvg    | 1.00  |   81.53%     |  82.66%      |   83.22%     |
>     | *FedLUAR*  | 0.17  |   81.88%     |   82.80%     |   82.75%     |
>
> In both benchmarks, *FedLUAR* achieves comparable accuracy to FedAvg regardless of $\alpha$, while considerably reducing the communication cost. Therefore, we conclude that ***FedLUAR* is robust to the degree of non-IIDness**.
>
> --
>
> **Q3**: What is the extra memory footprint at the server-side?
>
> **A3**: We would like to clarify that ***FedLUAR*’s memory footprint is actually smaller than that of FedAvg** (We realized this misanalysis after submitting the paper). Since all methods require the same amount of memory to store the global model, we focus only on the memory consumption of the local models.
>
>  - In FedAvg, server should receive local models from all active clients. Thus, the memory footprint is $a \cdot d$, where $a$ is the number of active clients and $d$ is the model size.
>  - *FedLUAR* receives local models from active clients except $\delta$ layers. Thus, the memory footprint is $a \cdot (d - k)$, where $k$ is the size of $\delta$ layers. Instead, the previous global update should be kept in the memory space for the $\delta$ layers, consuming $k$ space only. Therefore, *FedLUAR*’s memory footprint is $a \cdot (d - k) + k < a \cdot d$.
>
> To support our analysis, we actually measured the memory footprint of FedAvg and *FedLUAR* during training. First, the total number of clients is 128 and only randomly selected 32 clients are activated at each communication round. We use MPI to run FL on 2 GPUs. Thus, each process locally train 16 models and then all the locally trained models are aggregated using *MPI_Allreduce()*. Under this setting, we measured the memory footprint of each process, observed during training.
>
> | CIFAR-10 (ResNet20) | $\delta$ | Memory Footprint (MB) |
> | :--- | :---:|:---:|
> |FedAvg| - | 33.49 |
> | *FedLUAR* | 10 | 15.23 |
>
> | CIFAR-100 (WRN28-10) | $\delta$ |  Memory Footprint (MB) |
> | :--- | :---:|:---:|
> |FedAvg | - | 4462.80 |
> |*FedLUAR* | 14 | 2604.88|
>
> | FEMNIST (CNN) | $\delta$ |  Memory Footprint (MB) |
> | :--- | :---:|:---:|
> |FedAvg | - | 806.11 |
> |*FedLUAR* | 2 | 204.73 |
>
> | AG News (DistillBERT) | $\delta$ |  Memory Footprint (MB) |
> | :--- | :---:|:---:|
> |FedAvg | - | 8294.18 |
> |*FedLUAR* | 30 | 1825.42|
>
> We clearly see that *FedLUAR* actually consumes less memory space than FedAvg. Thank you again for pointing out this. We will add a brief discussion on this memory consumption in Section 3.2 and Appendix.
>
> --
>
> **Q4**: Is *FedLUAR* orthogonal to parameter quantization or sparsification?
>
> **A4**: Yes, our empirical study already shows that the proposed update recycling method is complementary to quantization. In Table 2, we compared the performance between with and without applying *LUAR* to FedPAQ which is a representative quantization-based federated learning method. E.g., FedPAQ’s communication cost is further reduced from $50$% to $33$% by recycling 10 layers’ updates without compromising CIFAR-10 (ResNet20) accuracy. This empirical result demonstrates that our method is well harmonized with quantization even in non-IID settings.
>
> In addition, we performed extra experiments to show how *FedLUAR* works with sparsification methods. We compare the benchmark performance between with and without applying *LUAR*.
>  - CIFAR-10 (ResNet20) with PruneFL, a sparsification-based FL method. The number of recycled layers, $\delta = 10$ out of $20$ layers in ResNet20.
> | Method | Accuracy | Comm |
> | :--- | :---: | :---: |
> | PruneFL | 56.76% | 0.69|
> | PruneFL + *LUAR* | 55.43% | 0.49 |
>
>  - FEMNIST (CNN) with PruneFL. The number of recycled layers, $\delta = 2$ out of $4$ layers in a CNN.
> | Method | Accuracy | Comm |
> | :--- | :---: | :---: |
> | PruneFL | 69.42% | 0.19|
> | PruneFL + *LUAR* | 69.11% | 0.11 |
>
> In both benchmarks, we see that our proposed update recycling method maintains the accuracy while further reducing the communication cost of PruneFL. So, we conclude that *FedLUAR* is complementary to sparsification methods.

---

> > ### Comment · Reviewer_21T7 · 2025-08-03
> >
> > I thank the authors for conducting additional experiments to address my concerns. I am satisfied with the response.

---

### Decision · Program_Chairs · 2025-09-17

**Decision:**

Accept (poster)

**Comment:**

This paper shows a novel approach to communication-efficient federated learning that recycles historical updates for low-priority layers based on a gradient-to-weight ratio metric. The idea of reusing past updates rather than dropping them is original. The paper has a theoretical convergence analysis and empirical validation across multiple standard datasets (although focusing on CNNs), showing significant improvements, and demonstrating compatibility with other FL optimizers. Reviewers noted the dependence on a manually tuned hyperparameter related to recycled layers, the absence of large-scale experiments, however overall the consensus is that the strengths outweigh the negatives. Overall, I recommend acceptance, as the paper presents a new and well-supported method.